# Tool-Augmented Spatiotemporal Reasoning for Streamlining Video Question Answering Task

**Sunqi Fan, Jiashuo Cui, Meng-Hao Guo, Shuojin Yang†**

BNRist, Department of Computer Science and Technology, Tsinghua University

`fansq20@mails.tsinghua.edu.cn`, `{gmh, yangshuojin}@tsinghua.edu.cn`

## Abstract

Video Question Answering (VideoQA) task serves as a critical playground for evaluating whether foundation models can effectively perceive, understand, and reason about dynamic real-world scenarios. However, existing Multimodal Large Language Models (MLLMs) struggle with simultaneously modeling spatial relationships within video frames and understanding the causal dynamics of temporal evolution on complex and reasoning-intensive VideoQA task. In this work, we equip MLLM with a comprehensive and extensible Video Toolkit, to enhance MLLM's spatiotemporal reasoning capabilities and ensure the harmony between the quantity and diversity of tools. To better control the tool invocation sequence and avoid toolchain shortcut issues, we propose a Spatiotemporal Reasoning Framework (STAR) that strategically schedules temporal and spatial tools, thereby progressively localizing the key area in the video. Our STAR framework enhances GPT-4o using lightweight tools, achieving an 8.2% gain on VideoMME and 4.6% on LongVideoBench. We believe that our proposed Video Toolkit and STAR framework make an important step towards building autonomous and intelligent video analysis assistants. The code is publicly available at `https://github.com/fansunqi/VideoTool`.

## 1 Introduction

> *"Man is a tool-using animal. Without tools he is nothing, with tools he is all."*
>
> *– Thomas Carlyle*

The key to the VideoQA task lies in the model's capacity to accurately perceive and reason over both the temporal logic and spatial layout of video content. This renders VideoQA a compelling arena for assessing whether foundation models can comprehend dynamic, multidimensional real-world scenarios in a manner akin to human cognition. With the rapid development of Large Language Models (LLMs), existing approaches to VideoQA can be broadly classified into two categories: Video-centric Large Language Models (Video-LLMs) [2, 15, 50, 88, 93] and Tool-augmented Large Language Models (Tool-augmented LLMs) [10, 82, 85].

Video-LLMs, represented by Qwen-VL [2], are typically required to process a large number of video frames, resulting in a redundant and inefficient pipeline. Tool-augmented LLMs, represented by DoreamonGPT [82], have several fundamental limitations: (1) **Unidimensional tool utilization**. Existing tool-augmented large language models primarily focus on the use of tools along a single dimension—either spatial or temporal, failing to simultaneously ensure the ability to model spatial relationships within video frames and to understand the causal dynamics of temporal evolution. This limitation hinders the model's capacity for deep understanding and reasoning. (2) **Unbalanced number and diversity of tools**. As we know, increasing the number of tools can help compensate for

---

†Corresponding author

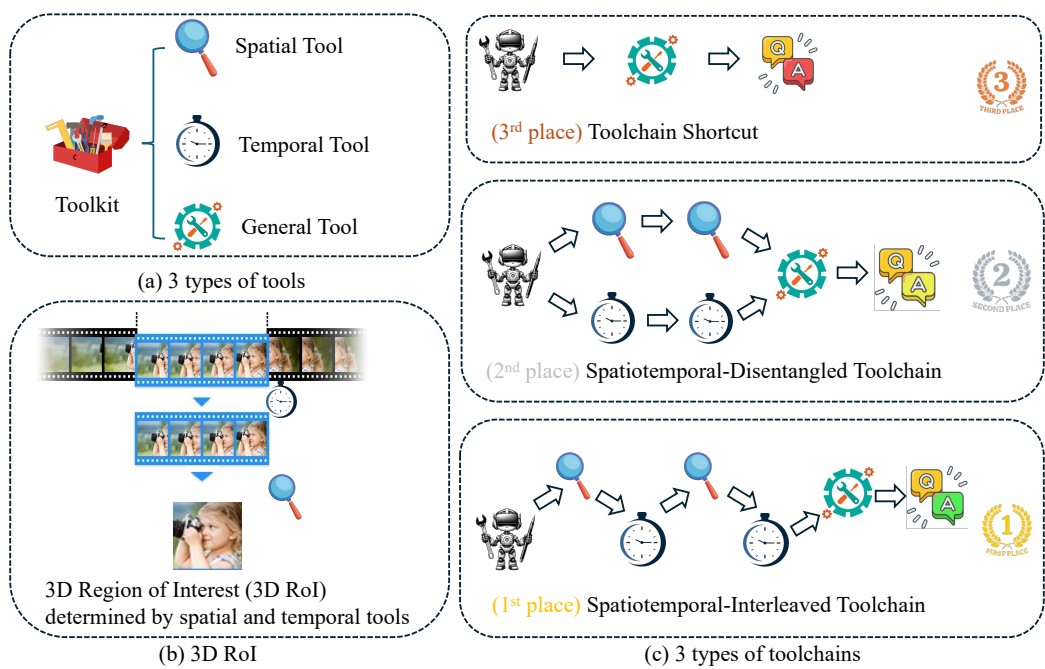

Figure 1: We categorize our toolkit into 3 types: spatial tools, temporal tools, and general tools. We also compare 3 toolchain strategies: the toolchain shortcut, which shows the lowest efficiency in tool utilization; the spatiotemporal-disentangled toolchain, which lacks mutual feedback between spatial and temporal reasoning; and the spatiotemporal-interleaved toolchain, which achieves the best accuracy and frame efficiency. We attribute this to its progressive localization of the 3D Region of Interest (3D RoI). See Section 4.2 for details.

many of the limitations present in LLMs. However, simply increasing the number of tools introduces various issues during execution, largely due to the inherent uncertainty of LLMs. (3) **Insufficient tool scheduling strategy**. The lack of an effective scheduling mechanism may lead to disordered tool invocation, which negatively affects the efficiency and accuracy of model reasoning.

To address the aforementioned limitations of Tool-augmented LLMs in previous works, we propose a new Video Toolkit to enhance MLLMs for VideoQA task. For example, by introducing tools with spatial localization capabilities—such as a lightweight object detection model [7], and combining them with basic operations like zooming and image cropping, we can effectively compensate for the shortcomings of MLLMs in spatial localization, thereby improving the accuracy of the VideoQA task. Additionally, tools can help filter out unimportant regions in both temporal and spatial dimensions, reducing the computational cost of image processing significantly.

Building on this video toolkit, we propose an iterative Spatiotemporal Reasoning Framework (STAR) to address the issues of insufficient strategy scheduling and avoid toolchain shortcut. The idea of this framework is to alternately narrow the scope along the temporal and spatial dimensions to locate the key regions that support the answer to the question. We refer to these regions as 3D Regions of Interest (3D RoIs). As shown in Figure 1, STAR generates spatiotemporal-interleaved toolchains, which facilitate the progressive localization of the 3D RoI to support question answering. STAR framework has the following features: (1) **Autonomy**. The LLM autonomously invokes tools within the framework. (2) **Adaptivity**. The framework dynamically adjust the tool invocation strategy based on the video's length, content, and the characteristics of the question. (3) **Progressiveness**. The framework progressively processes image frames, starting with a small number and gradually expanding as needed.

Overall, our contributions are as follows:

• We build a comprehensive toolkit specifically designed for video analysis to enhance spatiotemporal reasoning capabilities as well as ensure the harmony between the quantity and diversity of tools,

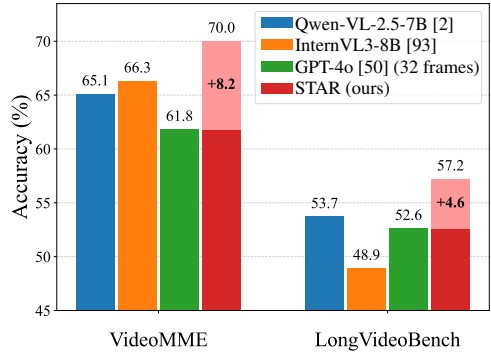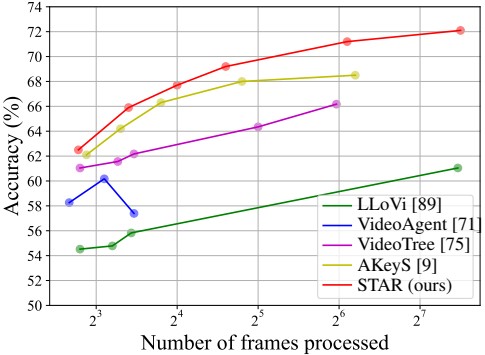

(a) Augmenting GPT-4o with Video Toolkit achieve an 8.2% gain on VideoMME [12] and 5% on LongVideoBench [77].

(b) As the number of input frames increases, STAR consistently achieves the highest accuracy on EgoSchema [44].

which includes 22 different tools covering a wide range of functionalities. All tools are plug-and-play, making this Video Toolkit highly extensible and easy to expand.

• We propose a novel reasoning framework named STAR that enables LLM to alternately invoke temporal and spatial tools. STAR facilitates stepwise visual reasoning by adapting to intermediate tool results, progressively narrowing down essential information to solve the problem.

• Through extensive experiments across multiple VideoQA benchmarks, we demonstrate that augmenting GPT-4o with lightweight Video Toolkit, results in an 8.2% gain on VideoMME [12] and 4.6% on LongVideoBench [77]. It also outperforms the current leading 7B Video-LLMs shown in Figure 2(a). We also show that STAR has stronger scalability shown in Figure 2(b).

## 2 Related Work

### 2.1 Video Question Answering

Video Question Answering (VideoQA) requires machines to answer questions based on videos. Driven by growing demands from real-world applications, the research community has progressively developed more sophisticated benchmarks [12, 18, 19, 30, 44, 54, 57, 64, 77, 78, 92], featuring longer videos and richer spatial and temporal dependencies. The emergence of transformer architectures has advanced video question answering through cross-modal pretraining and spatiotemporal modeling [4, 26, 39, 59, 69, 79, 91]. Recent advances in Video-LLMs [2, 5, 31, 32, 40, 68, 72, 73, 88, 93] have enabled open-ended video question answering by integrating LLMs with video encoders. Additionally, LLMs are used to reason across the temporal dimension and select keyframes adaptively for video understanding [20, 23, 24, 62, 71, 74, 75, 81, 85, 89]. Some other works [36, 51, 61, 86] improve computational efficiency by first using lightweight network modules to extract keyframes. [3, 11] builds scenes or event graphs to enable stepwise video understanding.

### 2.2 Tool Learning

Tool learning has emerged as a promising direction to enhance (M)LLMs by equipping them with the ability to invoke external tools. Early works focus on augmenting LLMs with API calling capabilities to perform tasks beyond their parametric knowledge, such as computation, web search, and code execution [45, 56, 83, 33, 34]. Integrating tools with LLMs also boosts their reasoning abilities, especially when multiple tools are combined in sequence to solve complex tasks [13, 27, 43]. Some works fine-tune LLMs for tool use, while others integrate tools in a plug-and-play manner without extra training. The order of tool invocation is crucial, and researchers have built large-scale tool network datasets [52, 58] to study how to better orchestrate sequential tool calls. Tool-augmented LLMs have also been applied to solve various tasks in computer vision by invoking specialized visual tools [14, 21, 25, 35, 37, 65]. While most existing studies focus on image-based applications, a growing number of works [8, 10, 29, 60, 82, 90] have explored video understanding through tool and

visual program augmentation. DoraemonGPT [82] relies on text-to-SQL queries over an intermediate database containing tools' outputs, which often fail and thus hurt both accuracy and efficiency. In contrast, our STAR framework circumvents this issue with a more reliable tool execution pipeline.

## 3 Method

### 3.1 Video Toolkit Construction

In this section, we first describe the design principles and construction process of the proposed Video Toolkit. The design of the Video Toolkit is guided by the following principles. (1) Video processing tasks naturally decompose into temporal and spatial dimensions. Accordingly, the Video Toolkit incorporates three types of tools: temporal, spatial, and general-purpose. (2) Video Toolkit should integrate a wide range of computer vision results through natural language interface. For example, how can outputs such as bounding boxes from object detection be integrated into natural language to assist in question answering? (3) In the temporal dimension, the Video Toolkit should support both segment-level and frame-level processing, enabling analysis of video segments as well as individual frames. Following these three design goals, we developed a comprehensive Video Toolkit consisting of about 22 video analysis tools. **The full list of tools and implementation details are provided in Appendix Section A**. Some of these tools are based on lightweight and specialized models, while others are simple image and video manipulations (e.g., the Patch Zoomer, which can zoom in or out on image regions). Additionally, to fully exploit the potential of MLLMs, we encapsulate several effective prompting pipelines as tools (e.g. iterative prompting MLLM for open-vocabulary action localization [67]). Below, we present two representative tools.

**Object Detection and Bounding Box Utilization.** Spatially, one of the most commonly used tools is object detection tool. To support different usage scenarios, we implemented two versions of object detection tool based on YOLO [7] and Grounding DINO [38], respectively. A typical tool chain involving object detection tool proceeds as follows: the LLM analyzes the question to determine the target object, which is then detected in the image using YOLO or Grounding DINO, yielding a set of Bounding Boxes (bboxes). We further implement three ways to utilize the resulting bboxes: (1) Converting the bboxes into textual descriptions and feeding them back to the LLM. (2) Using the Patch Zoomer tool to enlarge the regions within the bboxes. (3) Following Set-of-Mark Prompting [80], we overlay the bbox regions with visual markers to highlight them before passing the image back to the MLLM for further visual reasoning.

**Frame Selector for Temporal Navigation.** Temporally, the core tool is the Frame Selector, which is based on an LLM. It selects informative frames while discarding irrelevant ones based on the question and accumulated information. Initially, the Frame Selector collects sparsely and uniformly sampled frames. As more information is gathered through the use of other tools, we instruct the LLM to imagine the full video and reason about which temporal segments are most relevant to answering the question. The Frame Selector can output a single key frame, in which case the tool chain typically proceeds with spatial tools designed for image-level processing; it can also call a video trimming tool to extract segment-level video clips for subdivision. We also integrate several efficient (M)LLM-based keyframe selection pipelines. For example, AKeyS [9] considers both the question and scene transitions, and selects keyframes using an $A^*$-based search strategy. Another example is $T^*$ [85], which arranges multiple frames into an image grid and feeds it into the MLLM for selection.

Following Octotools [42], our Video Toolkit employs standardized tool cards to encapsulate each tool's functionality. All tools are designed to be plug-and-play, ensuring high extensibility and ease of integration.

### 3.2 Spatiotemporal Reasoning Framework

By augmenting (M)LLM with Video Toolkit, we propose STAR, a training-free, user-friendly, and extensible agentic and reasoning framework for streamlining VideoQA task across diverse domains.

**Toolchain Shortcut.** Our initial design allows the core LLM Planner to autonomously select tools from the entire toolkit and fully control the sequence of tool invocations (i.e., the toolchain [94]).

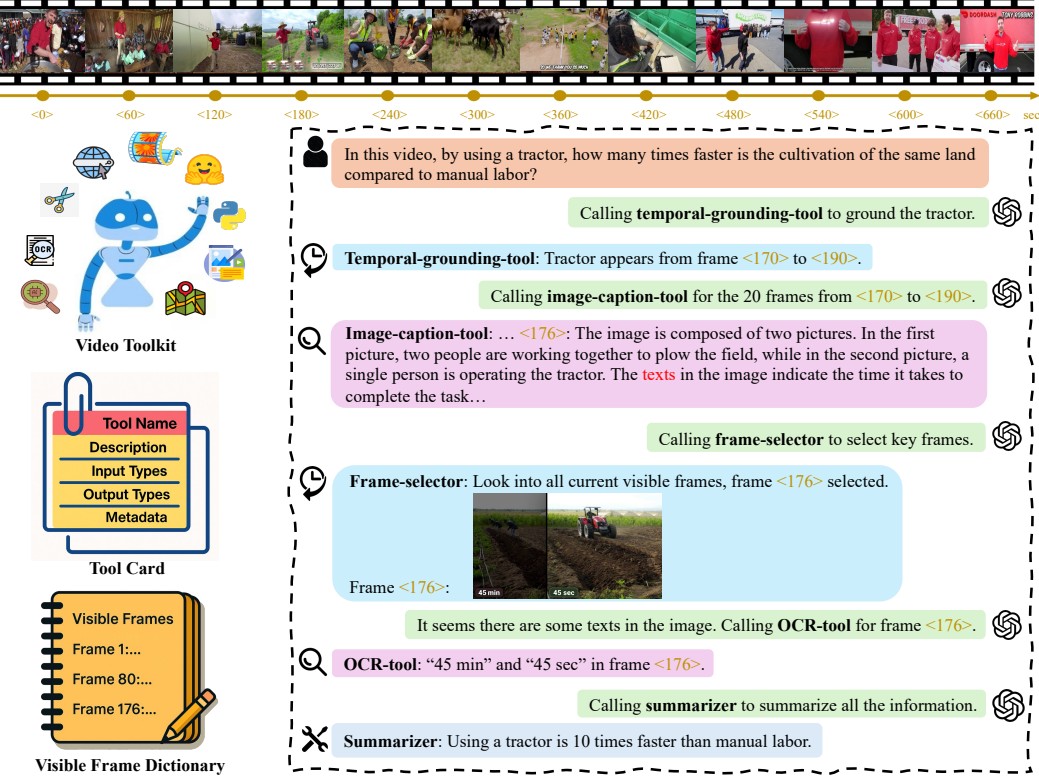

Figure 2: Visualization of our video toolkit, tool cards, and visible frame dictionary. Demonstration of the STAR pipeline. In this case, the LLM planner sequentially invokes five tools—temporal grounding, image captioning, frame selection, OCR, and summarization—to solve the problem.

Following the ReAct [83] agent framework, the LLM Planner makes an observation based on each tool invocation's results. The LLM Planner then generate a thought based on the observation, which decides the selection of the next tool to invoke or whether enough information has been gathered to answer the question. However, this loosely constrained and overly flexible design can sometimes lead to *Toolchain Shortcut* problem. We define *Toolchain Shortcut* as a phenomenon where the LLM Planner takes shortcuts by favoring single-step tools to directly answer the question, rather than progressively decomposing the problem and constructing a longer, step-by-step toolchain for reasoning. In our VideoQA task, toolchain shortcut manifests when the LLM Planner directly invokes a general-purpose tool (e.g., a video-language model) to answer the question, bypassing our intended strategy of progressively breaking down the problem. Our ablation study 4.2 offers a detailed examination of how Toolchain Shortcut affects both the accuracy and computational efficiency of the system.

**STAR Algorithm** To address Toolchain Shortcut problem, we impose a constraint on the toolchain: temporal and spatial tools must be invoked in an alternating manner, and general tools can only be called as a last resort when temporal and spatial tools alone are insufficient to answer the question. The initial tool can be either temporal or spatial, depending on the task. We also maintain a **Visible Frame Dictionary** $V$, where the keys are the indices of visible frames and the values are information gathered by various tools for each frame. At initialization, we populate this dictionary with a sparse set of uniformly sampled video frames, which initially contain no additional information. We then start the tool calling process, where temporal tools are responsible for selecting the frame indices to be processed—which can be a single frame, a continuous segment, or a set of discrete frames—and updating the visible frame dictionary by adding or removing frame indices as needed.

Spatial tools are responsible for processing the frames specified by the temporal tools and updating the corresponding values in the Visible Frame Dictionary $V$ for the respective frame indices. At each step, the core LLM Planner decides whether the information in the Visible Frame Dictionary is sufficient to answer the question. If not, it determines which tool to invoke next. If the first tool

**Algorithm 1** Spatiotemporal Reasoning (STAR)

---

**Require:** Video $v$, question $q$, LLM Planner $P$,
    iteration num $I$, Visible Frame Dictionary
    $D$, spatial toolset $T_s$, temporal toolset $T_t$,
    general toolset $T_g$

**Ensure:** Answer $\hat{y}$
1:  $D \leftarrow \text{UniformSparseSample}(v)$
2:  $t \leftarrow \text{SelectTool}(P, D, q, T_s \cup T_t)$
3:  $r \leftarrow \text{ExcuteTool}(v, t)$
4:  $D \leftarrow \text{UpdateDict}(D, r)$
5:  **for** $i = 1$ to $I$ **do**
6:     $T_{\text{next}} \leftarrow \begin{cases} T_t & \text{if } t \in T_s \\ T_s & \text{if } t \in T_t \end{cases}$
7:     $t \leftarrow \text{SelectTool}(P, D, q, T_{\text{next}})$
8:     $r \leftarrow \text{ExcuteTool}(v, t)$
9:     $D \leftarrow \text{UpdateDict}(D, r)$
10: **end for**
11: $t \leftarrow \text{SelectTool}(P, D, q, T_g)$
12: $\hat{y} \leftarrow \text{ExcuteTool}(D, t, v)$
13: **return** $\hat{y}$

---

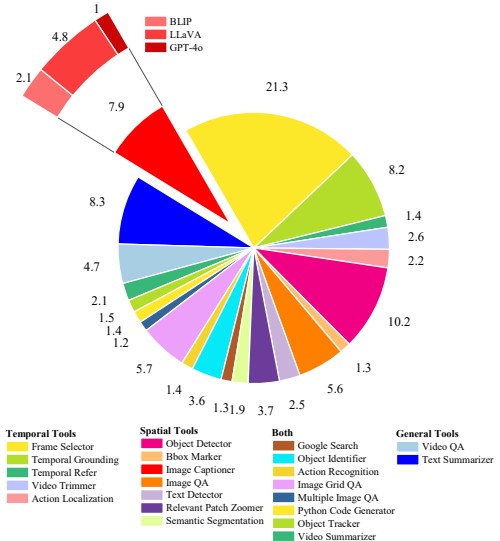

Figure 3: Diverse tool usage distribution on VideoMME, showcasing a wide range of tools being actively utilized.

invoked is a temporal tool, the LLM Planner will select tools from the temporal toolset at every odd step, and from the spatial toolset at every even step; the process works in reverse if the first tool is spatial. If the LLM Planner determines that neither temporal nor spatial tools can resolve the issue, it will resort to general tools to solve the problem. We conclude this algorithmic procedure in Figure 2 and Algorithm 1. We also visualize the percentage distribution of tool usage on the VideoMME [12] dataset in Figure 3, showing that our tools are utilized in a well-balanced way, with no tool being excessively favored or underused.

**Why spatiotemporal interleave?** Spatiotemporal tool interleaving ensures that, after temporal tools narrow the time range, spatial tools can further refine the spatial range. The results from spatial tools, in turn, influence the subsequent use of temporal tools, ensuring that the understanding of time and space is **complementary**. If time and space were decoupled, with time being progressively narrowed first and then space, without information transfer between the two, it would lead to a decline in both the accuracy and efficiency of video understanding, as demonstrated in our ablation study 4.2.

**How STAR mitigates the limitations of MLLMs in spatiotemporal reasoning?** STAR framework mitigates the spatiotemporal reasoning limitations of MLLMs mainly in two ways. (1) Specialized tools can generate more accurate outputs than MLLMs for specific tasks. For example, in a video with heavy pedestrian traffic, a Video-LLM often struggles with estimating the total number of unique individuals. In contrast, when the LLM invokes object tracking and person re-identification tools, the accuracy can be significantly improved. (2) Even when tools cannot directly produce definitive answers, the STAR framework progressively narrows down the temporal and spatial scope to localize the question-relevant 3D Region of Interest (3D RoI). In video analysis, a 3D RoI refers to a specific subset of the video defined across both spatial (height and width) and temporal (time) dimensions. By narrowing down and focusing on 3D RoI, STAR reduces irrelevant content and minimizes interference. Similar to how chain-of-thought prompting [76] elicits deliberate, system II reasoning [41] in complex reasoning tasks, our tool-augmented spatiotemporal reasoning framework also encourages such deliberate visual thinking, leading to improved accuracy and computational efficiency in complex video question answering scenarios.

## 4 Experiments

We evaluate our method on four widely used VideoQA datasets: VideoMME [12], NExT-QA [78], LongVideoBench [77], and EgoSchema [44]. These datasets cover diverse video lengths (ranging

Table 1: Results on VideoMME. We highlight the best results for each method category in bold, and mark our improvements over GPT-4o in blue.

| Model | Size | Frames ↓ | Runtime ↓ | Short ↑ | Medium ↑ | Long ↑ | All ↑ |
|---|---|---|---|---|---|---|---|
| *Proprietary Image-based MLLMs* | | | | | | | |
| GPT-4o [50] | - | 1 fps / 384 | > 10 min | 80.0 | 70.3 | 65.3 | 71.9 |
| GPT-4o | - | 32 | **< 30 s** | 68.3 | 60.7 | 56.3 | 61.8 |
| GPT-4o + $T^*$ [85] | - | 32 | 30 s - 1 min | 69.5 | 63.5 | 59.3 | 64.1 |
| GPT-4v [49] | - | **10** | **< 30 s** | 70.5 | 55.8 | 53.5 | 60.0 |
| Gemini 1.5 Pro [15] | - | 0.5 / 1 fps | > 10 min | **81.7** | **74.3** | **67.4** | **75.0** |
| Claude 3.5 Sonnet [1] | - | 20 | **< 30 s** | 71.0 | 57.4 | 51.2 | 60.9 |
| *Open-source Video-LLMs* | | | | | | | |
| Qwen2-VL [70] | 72B | 2 fps / 768 | 6 min - 8 min | **80.1** | **71.3** | 62.2 | **71.2** |
| Qwen2-VL | 7B | - | - | - | - | - | 63.3 |
| Qwen2.5-VL [2] | 7B | - | - | - | - | - | 65.1 |
| InternVL2.5 [6] | 8B | **64** | **< 30 s** | - | - | - | 64.2 |
| InternVL3 [93] | 8B | **64** | **< 30 s** | - | - | - | 66.3 |
| VideoLLaMA3 [88] | 7B | 180 | 30 s - 60 s | **80.1** | 63.7 | 54.9 | 66.2 |
| mPLUG-Owl3 [84] | 7B | 128 | 30 s - 60 s | 70.0 | 57.7 | 50.1 | 59.3 |
| STAR (ours) | - | **30.2** | **15.8 s** | 78.9 | 68.3 | 62.9 | **70.0 (8.2 ↑)** |

Table 2: Results on LongVideoBench Validation Set. We highlight the best results for each method category in bold, and mark our improvements over GPT-4o in blue.

| Model | Params | Frames ↓ | Runtime ↓ | 8s-15s ↑ | 15s-60s ↑ | 180s-600s ↑ | 900s-3600s ↑ | All ↑ |
|---|---|---|---|---|---|---|---|---|
| *Proprietary Image-based MLLMs* | | | | | | | | |
| GPT-4o [50] | - | 256 | > 10 min | **71.6** | **76.8** | **66.7** | **61.6** | **66.7** |
| GPT-4o | - | 32 | **< 30 s** | 60.7 | 62.4 | 50.1 | 49.0 | 52.6 |
| Gemini 1.5 Pro [15] | - | 256 | > 10 min | - | - | - | - | 64.0 |
| *Open-source Video-LLMs (~7B)* | | | | | | | | |
| Qwen2.5-VL | 7B | 1 fps / 512 | 1 min - 3 min | **59.0** | 65.6 | **52.9** | **48.8** | **53.7** |
| InternVL3 | 8B | 256 | 1 min - 3 min | 54.7 | **66.9** | 46.1 | 44.0 | 48.9 |
| STAR | - | **29.6** | **15.3 s** | 63.6 | 68.0 | 56.8 | 52.3 | **57.2 (4.6 ↑)** |

from a few seconds to several hours), video types and question types, enabling a comprehensive comparison between our approach and baseline methods. Introductions of these datasets are provided in Appendix B. We primarily compare our method against the following four categories of baselines: (1) image-based MLLMs, (2) Video-LLMs, (3) (M)LLM-based frame selection methods, and (4) LLM-driven tool learning methods. Detailed introductions of these baselines can be found in Appendix C. We use multiple-choice **QA accuracy** to measure performance, and use **the number of processed frames** and **runtime** (the average time to answer a question under identical conditions) to measure computational efficiency.

To accommodate different computational budgets and compare fairly, we design two variants of our framework depending on whether the full video toolkit is used: STAR and STAR-MINI. STAR is the full version, employing tools based on open-source models with up to 3B parameters (e.g., QwenVL-2.5-3B [2]) and using GPT-4o-2024-08-06 [50] APIs. Following previous baseline [9], It uses the same version GPT-4o as its LLM Planner. STAR-MINI excludes any tools larger than 500M parameters; its largest tool is a 500M-parameter BLIP [28]. It also avoids any tools that require GPT-4o APIs. Following previous baseline [82], It uses the same version GPT-3.5-turbo-0125 [48] as its LLM Planner. Our STAR framework can run on one NVIDIA RTX 4090 GPU, while the variant STAR-MINI framework can run on a personal computer like MAC. For videos longer than 16 seconds, we extract 16 initial frames uniformly; for videos with a duration of 16 seconds or less, initial frames are extracted at a rate of 1 fps.

Table 3: Results on NExT-QA Test Set. We highlight the best results for each method category in bold, and mark our improvements over the best baseline $T^*$ in blue.

| Method | Base Model | Frames ↓ | Cau. ↑ | Tem. ↑ | Des. ↑ | All ↑ |
|---|---|---|---|---|---|---|
| *Open-source Video-LLMs (~7B)* | | | | | | |
| InternVL3-8B [93] | - | **8** | 77.5 | 72.1 | 77.1 | 75.7 |
| Qwen2.5-VL-7B [2] | - | 2 fps | **80.6** | **79.3** | **85.5** | **80.9** |
| *(M)LLM-based Frame Selection Methods* | | | | | | |
| VideoAgent [71] | GPT-4 | 8.2 | 64.5 | 72.7 | 81.1 | 71.3 |
| VideoTree [75] | GPT-4 | 63.2 | 70.6 | 76.5 | 83.9 | 75.6 |
| LVNet [51] | GPT-4o | 12 | 65.5 | 75.0 | 81.5 | 72.9 |
| VidF4 [36] | - | 8 | 69.6 | 74.2 | 83.3 | 74.1 |
| AKeyS [9] | GPT-4o | **7.6** | **72.9** | **79.0** | **86.1** | 78.1 |
| Detector-based $T^*$ [85] | GPT-4o | 8 | - | - | - | **80.4** |
| STAR (ours) | GPT-4o | **7.2** | **81.1** | **81.5** | **86.3** | **82.1 (1.2 ↑)** |

Table 4: Results on NExT-QA Validation Set. We highlight the best results for each method category in bold, and mark our improvements over the best baseline DoraemonGPT [82] in blue.

| Method | Base Model | Frames ↓ | #LLM calls ↓ | Cau. ↑ | Tem. ↑ | Des. ↑ | All ↑ |
|---|---|---|---|---|---|---|---|
| *LLM-driven Tool Learning Methods* | | | | | | | |
| ViperGPT [60] | GPT-3 Codex | - | - | 43.2 | 41.0 | 62.3 | 45.5 |
| VideoChat [29] | - | - | - | 50.2 | 47.0 | 65.7 | 51.8 |
| DoraemonGPT [82] | GPT-3.5-turbo | 28.7 fps / 1144.4 | 8.5 | **54.7** | **50.4** | **70.3** | **55.7** |
| STAR-MINI (ours) | GPT-3.5-turbo | **0.6 fps / 22.6** | **5.4 (3.1 ↓)** | **62.8** | **55.1** | **73.7** | **62.0 (6.3 ↑)** |

## 4.1 Main Results

### 4.1.1 Results on VideoMME

As shown in Table 1, with the nearly same number of input frames, our method achieves an 8.2% improvement over GPT-4o [50]. Meanwhile, our method significantly outperforms all open-source Video-LLMs around 7B scale (achieving a 3.7% improvement over the best-performing InternVL3-8B), and approaches the performance of open-source Video-LLMs around 72B scale. Moreover, in terms of efficiency, our method substantially reduces the number of video frames required for processing. Compared to the 72B Qwen2-VL, our method achieves a substantial speedup, reducing runtime from 6–8 minutes to 15.8 seconds. We reproduce GPT-4o with 32 input frames and other baseline results in Table 1 come from the official leaderboard and technical reports.

### 4.1.2 Results on LongVideoBench

As shown in Table 2, with the nearly same number of input frames, our method achieves a 4.6% improvement over GPT-4o. Meanwhile, our STAR framework outperforms both Qwen2.5-VL-7B and InternVL3-8B by a large margin, particularly in the long (180–600 s) and extra-long (900–3600 s) parts. This demonstrates that our method is more effective than the current 7B Video-LLMs at identifying key information from long videos. Moreover, the STAR framework substantially reduces both the number of processed frames and runtime, leading to lower computational cost. We reproduce GPT-4o's results with 32 input frames using no subtitle information. We also reproduce the performance of Qwen2.5-VL-7B and InternVL3-8B on the validation set without using subtitle information for comparison. Results of Gemini and GPT-4o with 256 input frames are taken from the official leaderboard. Their use of subtitle information is one of the key reasons why they outperform the STAR framework. STAR currently focuses solely on the video content; we consider better integration of subtitle and audio information as future work.

### 4.1.3 Results on NExT-QA

As shown in Table 3, STAR achieves the highest accuracy while using the fewest frames, outperforming both 7B Video-LLMs and all (M)LLM-based frame selection methods. It attains over 80%

Table 5: Ablation on Different Methods to Generate Toolchain

| Methods | Accuracy ↑ | Num of Frames ↓ | Toolchain Length ↑ | Num of Tools ↑ |
|---|---|---|---|---|
| **No Constraints** | 61.2 | 112.6 | 2.9 | 1.3 |
| **Prompting** | 60.4 | 98.7 | 3.6 | 1.9 |
| **In-Context Learning** | 63.2 | 50.1 | 5.4 | 3.2 |
| **Spatiotemporal Disentanglement** | **68.6** | **40.6** | **5.6** | **3.4** |
| **STAR** | **70.0 (1.4 ↑)** | **30.2 (10.4 ↓)** | **8.7 (3.1 ↑)** | **6.3 (2.9 ↑)** |

accuracy across all three question categories—causal, temporal, and descriptive—demonstrating its effectiveness across diverse question types. The performance of open-source Video-LLMs is our own reproduction, and the results of (M)LLM-based frame selection methods are reported from their original papers. And in Table 4, on the NExT-QA validation set, STAR-MINI outperforms Doraemon-GPT [82] by 6.3% in accuracy, demonstrating the superiority of the spatiotemporal interleaving tool invocation strategy. All the results of LLM-driven tool learning methods are reported from their original papers.

## 4.2 Ablation Studies

**Ablation on Toolchain**    Besides the STAR framework, we also explore the following strategies to control the tool invocation order. (1) No Constraints: The LLM Planner has no restrictions; all tool information is provided in tool cards, and the planner autonomously selects tools according to the question and the tool cards. In practice, this often leads to the problem of ***Toolchain Shortcut***. (2) Prompting: Inspired by chain-of-thought prompting [76], we instruct the LLM Planner to invoke longer and more complex toolchains, solving the problem step by step. (3) In-Context Learning: We provide manually annotated examples with long toolchains, leveraging the LLM's in-context learning ability to imitate and generate more complex tool sequences. (4) Spatiotemporal Disentanglement: The LLM Planner is guided to first invoke one or more temporal tools to narrow the temporal scope, followed by one or more spatial tools to refine the spatial scope, and finally either directly generate the answer or optionally call general tools before answering.

We conducted experiments on VideoMME [12] to compare these strategies in Table 5. The No Constraints method results in overly short toolchains and repetitive tool usage, leading to excessive frame processing and high computational costs. The Prompting method has only marginal improvements over the No Constraints method, indicating that prompt-based control alone is insufficient. The In-Context Learning method provides stronger control compared to the Prompting method, significantly reducing the number of frames processed and increasing toolchain length. However, it still falls short in terms of accuracy. Both Spatiotemporal Disentanglement and STAR framework impose explicit constraints on the LLM Planner by directly controlling the toolchain structure. Among them, the spatiotemporal-interleaved STAR achieves the best accuracy and frame efficiency, benefiting from longer, more diverse toolchains. We mark STAR's improvements over the second-best Spatiotemporal Disentanglement method in blue.

**More Ablations and Analysis**    We also conducted ablation study on each tool, showing that each tool contributes positively to the overall performance in Appendix Section D. Additionally, Appendix Section E examines the impact of varying frame numbers on the performance of the STAR framework, while Appendix Section F investigates the effects of different LLM Planners on its performance. For more in-depth analysis, we verified whether the STAR framework promotes a balanced and comprehensive use of tools, as presented in Appendix Section G. We also conduct more case studies in Appendix Section I and summarize the failure cases in Appendix Section H.

## 5 Conclusion

In this work, we equip (M)LLMs with a carefully designed and comprehensive video toolkit to compensate for the temporal and spatial reasoning limitations of MLLMs, streamlining the VideoQA task. To better control tool invocation, we propose a Spatiotemporal Reasoning framework (STAR), which alternately calls temporal and spatial tools to progressively localize the 3D RoI, thereby improving accuracy and reducing computational cost. We conduct extensive experiments on four

widely used VideoQA datasets. Our STAR framework, built with lightweight models of fewer than 3B parameters, significantly boosts GPT-4o's ability in video understanding and reasoning, and consistently outperforms related baselines in accuracy and efficiency.

Our main limitation lies in that STAR framework still relies on the capabilities of OpenAI GPT models, which can lead to certain API costs due to repeated invocations. In future work, we plan to explore replacing GPT-4o planner with more lightweight models and investigate techniques to better integrate tool usage into the reasoning process of the planner. Nevertheless, we believe that our proposed Video Toolkit and STAR framework make an important step towards building autonomous and intelligent video analysis assistants.

**Acknowledgement**   This research was supported by the National Natural Science Foundation of China (project No. 62495060, 623B2057), the Research Grant of Tsinghua-Tencent Joint Laboratory for Internet Innovation Technology.

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

# Appendix

## A Tool Detail

### A.1 Temporal Tools

**Frame Selector** Refer to Section 3.1 of the main text. Specifically, we implement 3 variants of the frame selector for different computational resources.

- Vanilla version: The frame selector is an LLM that takes as input the IDs and textual descriptions of all visible frames, such as captions generated by an image captioner. It outputs the selected frame IDs. The prompt is directly instructing the LLM to select relevant frames.
- AKeyS-inspired version: This variant shares the same input and output format as the Vanilla version but employs specially designed prompts [9] to encourage the LLM to consider task goals and scene transition.
- T*-inspired version: In this case, the frame selector is an MLLM, and its input additionally includes the actual images of visible frames. These images are further arranged into an image grid to facilitate selection [85]. The output remains the same.

The selected frame IDs produced by the frame selector are subsequently added to the Visible Frame Dictionary.

**Temporal Grounding Tool** We adopt the most lightweight version of Grounded-VideoLLM [68] (with approximately 4B parameters), which augments the language model with temporal tokens and is trained on a large corpus of temporally annotated video-text pairs. Given an event description, the tool predicts its temporal span in the video.

**Temporal Referring Tool** For temporal referring, we use the same model as in temporal grounding, but with a different input-output format. Instead of grounding a described event in time, the tool takes a temporal span as input and generates a textual description of the visual content within that interval.

**Video Trimmer** Video Trimmer is a Python-based tool to trim the video along the temporal axis based on a temporal span.

**Action Localization Tool** We implement an action localization tool based on the method proposed in [66]. The core idea is to organize the video into an image grid and feed it into GPT-4o [50], which then identifies the frame indices corresponding to a specified action.

### A.2 Spatial Tools

**Object Detector** Refer to Section 3.1 of the main text.

**Bbox Marker** Following Set-of-Mark [80], we annotate the 2D Regions of Interest (RoIs) on corresponding frames, which enhances the effectiveness of visual question answering by MLLMs such as GPT-4o.

**Image Captioner** We employ three Vision-Language Models (VLMs) of different scales to perform the image captioning task: BLIP2 (about 1B parameters), LLaVA (about 7B parameters), and GPT-4o. The input to this tool is individual video frames, and the output is a textual description of each frame. These descriptions are stored and maintained in the Visible Frame Dictionary.

**Image QA Tool** We use the same models as those in the image captioning tool, but the input now consists of images paired with corresponding questions, and the output is textual answers. These answers are stored and maintained in the Visible Frame Dictionary.

**Text Detector** We use an off-the-shelf OCR tool to recognize text in video frames.

**Relevant Patch Zoomer**    Following Octotools [42], this tool is used to enlarge the RoI in an image and crop out irrelevant areas before further processing.

**Semantic Segmentation Tool**    We use Grounded-SAM [55] model for semantic segmentation.

### A.3    Tools that can be regarded as both temporal and spatial tools.

**Google Search**    When faced with questions that exceed its internal world knowledge, the LLM Planner can invoke the Google Search API to retrieve relevant external information, thereby enabling knowledge-enhanced VideoQA.

**Object Identifier**    Inspired by $T^*$ [85], this tool is designed to identify key objects mentioned in the question, serving as a preparatory step for subsequent object detection and tracking operations.

**Action Recognition Tool**    Action recognition involves understanding both the temporal dynamics and spatial cues of a video. We implement our action recognition tool based on MMAction [47], enabling the identification of target actions within the video.

**Image Grid QA Tool**    Following [85], we organize the visible frames of the video into an image grid, which is then treated as a single image for subsequent image question answering.

**Multiple Image QA Tool**    Instead of creating an image grid, we feed the visible video frames sequentially as individual images into the MLLM model for question answering.

**Python Code Generator**    This tool can generate and excute any Python code to manipulate videos.

**Object Tracker**    The object tracker can track specified objects in videos and includes person re-identification function. Our implementation is based on Ultralytics [63].

**Video Summarizer**    This tool employ a video-language model (Qwen2.5-VL-3B [2]) to perform captioning or summarization over the entire video, generating textual descriptions or summarizations.

### A.4    General-Purpose Tools

**Text Summarizer**    The textual outputs from each tool are maintained in the Visible Frame Dictionary for the corresponding frames. This tool is responsible for summarizing the information and attempting to answer the question.

**Video QA Tool**    This tool leverages a video-language model (Qwen2.5-VL-3B [2]) to directly perform video question answering, generating answers based on the input question and video content.

## B    Datasets

**VideoMME**    VideoMME [12] is the most widely adopted benchmark for evaluating video analysis capabilities. It is a test-only dataset, containing 2,700 multiple-choice VideoQA examples. VideoMME is known for its diversity and comprehensive coverage across video lengths, video domains, and question types. In terms of length, the dataset categorizes videos into short (<2 minutes), medium (4–15 minutes), and long (30–60 minutes) segments. Regarding video types, VideoMME spans nearly all categories commonly found on the internet, grouped into six domains: Knowledge, Film & Television, Sports Competition, Artistic Performance, Life Record, and Multilingual. Due to its broad coverage, VideoMME serves as an effective benchmark to assess whether a VideoQA system can perform robustly across different video lengths and domains, thus providing a solid measure of the system's generality and transferability.

**NExT-QA**    NExT-QA [78] dataset comprises 5,440 videos and approximately 52,000 human-annotated QA pairs. It is specifically curated to benchmark models' capabilities in understanding the causal structures and temporal dependencies inherent in complex video events. In this work, we

adopt the multiple-choice question answering subset of NExT-QA, which includes 34K training, 5K validation, and 9K testing samples. A notable characteristic of NExT-QA is its diverse question formulations, spanning various interrogatives including how, who, when, where, what, and why, thereby providing a comprehensive testbed for evaluating a model's adaptability to different query types. The questions are further categorized into three classes: causal questions that probe reasoning over cause-effect relations, temporal questions that assess understanding of event chronology, and descriptive questions that focus on visual and contextual details. Additionally, the videos in NExT-QA are relatively short, with an average duration of approximately 0.7 minutes.

**LongVideoBench**  LongVideoBench [77] is specifically designed to evaluate long-form video understanding, with videos averaging around 8 minutes in duration. Analogous to the "needle-in-a-haystack" challenge [46], LongVideoBench introduces referring reasoning questions, where each query references specific video contexts to guide the answering process. This formulation imposes greater demands on models' temporal reasoning capabilities. In this work, we evaluate our method on the validation split of LongVideoBench, which contains 1,337 questions. For fair comparison, we exclude the use of subtitle information provided by LongVideoBench and focus solely on video content understanding.

**EgoSchema**  The EgoSchema [44] dataset includes over 5,000 carefully curated multiple-choice question-answer pairs, making it a prominent benchmark for long-form video question answering. EgoSchema stands out for its challenging nature—human performance reaches only 76% accuracy, while current Video-LLMs fall below 70%. The dataset's extended video durations and high complexity highlight the critical need for key information retrieval.

## C  Baselines

Our evaluation focuses on a comparison between our method and the following four baseline approaches.

**Image-based MLLMs**  Image-based MLLMs address VideoQA tasks by converting videos into image sequences and guiding the models with carefully designed prompts. Thanks to their strong general capabilities, models like GPT-4o [50] and Gemini-2.5-Pro [16] have achieved impressive results on video question answering benchmarks such as VideoMME [12]. However, this approach has several limitations: (1) the best-performing image-based MLLMs are mostly proprietary and expensive to access via APIs, especially when handling long videos with extended context; (2) as discussed above, MLLMs exhibit limited spatiotemporal reasoning abilities, which require specialized lightweight tools for effective augmentation; and (3) their performance tends to degrade when the number of input frames is insufficient.

**Video-LLMs**  Video-LLMs are a specialized class of MLLMs designed for video-language understanding tasks. These models either incorporate video-specific architectural designs [5, 32, 87], are trained on large-scale video-text datasets [72], or include temporal modeling optimizations [31, 68]. In this study, we primarily examine the following representative Video-LLMs: Qwen-VL [2], InternVL [93] and VideoLLaMA [88]. Similar to general LLMs, Video-LLMs come in various parameter scales; in our experiments, we focus on models around 7B and 72B parameters. Our goal is for the STAR framework to outperform the 7B models and approach the performance of the 72B models. The performance of Video-LLMs is also influenced by the number of input frames provided.

**(M)LLM-based Frame Selection Methods**  Compared with this line of methods, our primary goal is to highlight the complementary role of our video toolkit in augmenting MLLMs. We include several representative works in this category. VideoAgent [71] first performs sparse and uniform sampling over the video and then leverages an LLM to dive deeper into important segments. VideoTree [75] builds a video tree structure guided by CLIP based on the given question, followed by a tree-based search using an LLM. LVNet [51] similarly constructs a hierarchical keyframe selector with CLIP, selecting key frames before feeding them into an MLLM for processing. VidF4 [36] proposes a differentiable adaptive frame sampling mechanism to enable end-to-end training of the frame selector. T* [85] and AKeyS [9] are directly integrated as tools within our video toolkit, as described earlier in

the Method section. We mainly compare our approach against these methods on the NExT-QA [78] dataset.

**LLM-driven Tool Learning Methods**    Several recent works have begun integrating tool learning into video understanding. ViperGPT [60] analyzes images and videos by generating and executing Python programs. VideoChat [29] incorporates various perception tools, such as Intern-Video [73], Whisper [53], Tag2Text [22], to perform multi-modal video analysis. Among them, DoraemonGPT [82] serves as our most important reference. It leverages tools like object trackers and BLIP [28] to pre-process videos, constructing space-dominant and time-dominant memory and then employs a text-to-SQL querying tool to retrieve answers from the memory. It uses Monte Carlo Tree Search (MCTS) to search for the best toolchain. To ensure a fair comparison, we built STAR-MINI using tools of comparable scale, and conducted experiments on the NExT-QA dataset against these methods.

## D    Ablation on each tool

We conducted ablation studies on each individual tool, demonstrating that every tool contributes positively to the overall performance. Keeping all other conditions unchanged, we remove a specific tool from the Video Toolkit and evaluate how its removal affects the accuracy of our method and the number of video frames processed. We conduct experiments on VideoMME [12] and LongVideoBench [77], recording the accuracy drop and frame increase caused by removing each individual tool, as shown in Table 6. We highlight the largest change within each tool category in bold, indicating the relative importance of that tool within the toolkit.

We observe that, in most cases, removing a tool leads to a drop in accuracy and an increase in the number of processed frames. In a few cases, tools that require many frames for computation result in reduced accuracy but fewer frames when removed. This indicates that nearly all tools contribute positively to the overall performance of our method.

## E    Scalability with more frames

We conducted experiments to evaluate the impact of increased frame sampling rates. Due to API budget and time constraints, we randomly sampled 1,000 examples from the Video-MME test set for this evaluation. The accuracy results are show in Table 7. They demonstrates that the STAR framework continues to improve performance as the number of sampled frames increases. For example, under a denser sampling setting (1 fps, up to 384 frames), the framework achieves a 5.2% accuracy gain.

## F    Generalizability with different base LLMs

We also examined the performance of the STAR framework when using different LLMs as the LLM Planner. The results in Table 8 show that, compared to their tool-free counterparts, these models generally demonstrate a 7%–8% improvement in accuracy, highlighting STAR effectiveness across model types.

## G    Analysis on Tool Balance

The STAR framework addresses the issue of unbalanced tool quantity and diversity primarily by removing the tools that the LLM Planner tends to over-rely on under the no-constraints setting (e.g., the Video QA Tool and Image QA Tool). With these tools no longer "monopolizing" the calls, the distribution of tool usage becomes more balanced both across and within tool categories.

In Table 9, we calculated the percentage of tool usage frequency on the Video-MME test set for both the No Constraints setting and the STAR framework. Based on Table 9, we compute the variance of tool usage counts within each tool category in Table 10. It can be observed that after adopting the STAR framework, the variance of tool usage counts within each category generally decreases, indicating a more balanced utilization of tools within each class.

Table 6: Performance Change on VideoMME and LongVideoBench after Removing Each Tool

| Tool Removed | VideoMME | | LongVideoBench | |
|---|---|---|---|---|
| | Acc Drop | Frames Increase | Acc Drop | Frames Increase |
| *Temporal Tools* | | | | |
| Frame Selector | **4.6** | **14.3** | **3.4** | **15.6** |
| Temporal Grounding Tool | 0.9 | 3.1 | 0.6 | 2.3 |
| Temporal Referring Tool | 0.8 | 2.9 | 0.3 | 1.2 |
| Video Trimmer | 0.8 | 11.5 | 0.9 | 7.2 |
| Action Localization Tool | 0.6 | 1.2 | 0.2 | 0.7 |
| *Spatial Tools* | | | | |
| Object Detector | 1.4 | 3.5 | 1.5 | **4.4** |
| Bbox Marker | 0.8 | 2.1 | 0.2 | 1.0 |
| Image Captioner | 1.3 | 2.4 | **1.6** | 3.5 |
| Image QA Tool | **1.5** | **5.5** | 1.2 | 3.5 |
| Text Detector | 0.4 | 0.3 | 1.1 | 2.1 |
| Relevant Patch Zoomer | 1.0 | 2.3 | 0.8 | 2.8 |
| Semantic Segmentation Tool | 0.3 | 0.7 | 0.1 | 0.4 |
| *Both Spatial and Temporal Tools* | | | | |
| Google Search | 0.3 | 0.5 | 0.2 | 0.4 |
| Object Identifier | 0.5 | 1.1 | 0.6 | 1.3 |
| Action Recognition Tool | 0.3 | 0.9 | 0.2 | 0.5 |
| Image Grid QA Tool | **1.2** | **6.8** | 1.5 | 7.8 |
| Multiple Image QA Tool | 0.9 | -0.1 | 1.0 | -0.4 |
| Python Code Generator | 0.2 | 0.0 | 0.0 | 0.3 |
| Object Tracker | 0.3 | 0.1 | 0.2 | 0.4 |
| *General Tools* | | | | |
| Text Summarizer | 0.9 | **0.2** | **1.3** | **1.1** |
| Video Summarizer | 1.0 | -0.3 | 0.7 | -0.2 |
| Video QA Tool | **2.3** | -0.2 | 1.1 | -0.4 |

Table 7: Performance with Different Frame Sampling Rates

| Model | LLM Planner | Avg. Frames | Short (%) | Medium (%) | Long (%) | All (%) |
|---|---|---|---|---|---|---|
| GPT-4o | - | 32 | 68.6 | 60.6 | 56.0 | 61.5 |
| GPT-4o | - | 100 | 72.8 | 63.2 | 59.5 | 64.9 |
| GPT-4o | - | 1 fps / 384 | 79.2 | 70.4 | 66.5 | 71.8 |
| STAR | GPT-4o | 31.3 | 78.2 | 68.7 | 62.7 | 69.6 (**+8.1%**) |
| STAR | GPT-4o | 100.2 | 80.1 | 71.0 | 66.4 | 72.4 (**+7.5%**) |
| STAR | GPT-4o | 0.98 fps / 384 | 85.3 | 78.0 | 68.5 | 77.0 (**+5.2%**) |

## H  Failure Cases

We analyzed the failure cases of the STAR framework and categorized the common ones into three major types:

- **Missing or ambiguous visual information.** In some cases, the video alone does not provide sufficient clarity and must be supplemented with subtitles or audio cues. For example, in Video-MME test set question 302-1, an arrow points to a character labeled "Victor Garber". However, this label does not imply that the character is Victor Garber; instead, it means that the actor Victor Garber plays this character. Understanding this requires reference to the narration. We plan to address such issues by incorporating subtitle inputs and developing tools for audio understanding in the future work.

- **Incomplete understanding of the video's main theme.** Our STAR framework sometimes fails to fully capture the primary focus of a video due to sparse frame sampling. For instance,

Table 8: Performance Comparison across Different LLM Planners

| Model | LLM Planner | Avg. Frames | Short (%) | Medium (%) | Long (%) | All (%) |
|---|---|---|---|---|---|---|
| GPT-4o [50] | - | 32 | 68.6 | 60.6 | 56.0 | 61.5 |
| Gemini-2.5-pro [16] | - | 32 | 77.6 | 62.6 | 57.1 | 65.4 |
| Qwen2.5-VL-72B [2] | - | 32 | 68.9 | 58.8 | 55.4 | 60.8 |
| STAR | GPT-4o | 31.3 | 78.2 | 68.7 | 62.7 | 69.6 (+8.1 ↑) |
| STAR | Gemini-2.5-pro | 31.0 | 79.8 | 70.7 | 69.1 | 72.9 (+7.5 ↑) |
| STAR | Qwen2.5-VL-72B | 31.5 | 77.9 | 66.1 | 62.4 | 68.5 (+7.7 ↑) |
| STAR | DeepSeek-R1 [17] | 31.2 | 77.2 | 67.2 | 63.0 | 68.9 |

Table 9: Tool Usage Frequency Comparison between No Constraints and STAR Framework

| Tool Type & Name | No Constraints (%) | STAR Framework (%) |
|---|---|---|
| **Temporal Tools All** | **32.1** | **35.7** |
| Frame Selector | 27.1 | 21.3 |
| Temporal Grounding | 2.1 | 8.2 |
| Temporal Referring | 0.0 | 1.4 |
| Video Trimmer | 1.3 | 2.6 |
| Action Localization | 1.6 | 2.2 |
| **Spatial Tools All** | **23.6** | **33.1** |
| Object Detector | 2.3 | 10.2 |
| Bbox Marker | 0.2 | 1.3 |
| Image Captioner | 2.1 | 7.9 |
| Image QA | 17.8 | 5.6 |
| Text Detector | 1.0 | 2.5 |
| Relevant Patch Zoomer | 0.2 | 3.7 |
| Semantic Segmentation | 0.0 | 1.9 |
| **Both (Temporal and Spatial Tools) All** | **5.4** | **16.1** |
| Google Search | 0.3 | 1.3 |
| Object Identifier | 1.0 | 3.6 |
| Action Recognition | 0.5 | 1.4 |
| Image Grid QA | 0.1 | 5.7 |
| Multiple Image QA | 2.8 | 1.2 |
| Python_Code_Generator | 0.0 | 1.4 |
| Object Tracker | 0.7 | 1.5 |
| **General Tools All** | **38.9** | **15.1** |
| Text Summarizer | 2.2 | 8.3 |
| Video Summarizer | 3.5 | 2.1 |
| Video QA | 33.2 | 4.7 |

in Video-MME test set question 303-3, the question is "What is the primary focus of this video for the audience?" but the sampled frames did not sufficiently reflect the video's theme. We aim to improve performance on such global reasoning tasks by encouraging denser sampling in future versions.

- **Difficulty in inferring underlying motivations behind human actions.** STAR struggles with questions that require understanding the deeper intent behind actions. For example, in Video-MME test set question 312-2, the question "Why does Michael Bierut mention religious symbols?" reveals challenges in modeling causal and temporal dependencies. This suggests that capturing fine-grained causal reasoning remains a fundamental difficulty in video understanding.

# I  Case Study

In Figures 4, 5 and 6, we present STAR's cases of diverse question types tackled by leveraging different tools from our Video Toolkit.

Table 10: Variance Comparison of Tool Usage between No Constraints and STAR Framework

| Tool Type | Var. of No Constraints | Var. of STAR Framework |
|---|---|---|
| Temporal Tools | 134.26 | 69.90 (**64.36** ↓) |
| Spatial Tools | 41.34 | 11.09 (**30.25** ↓) |
| Both | 0.92 | 2.95 (**2.03** ↑) |
| General Tools | 307.45 | 9.69 (**297.76** ↓) |

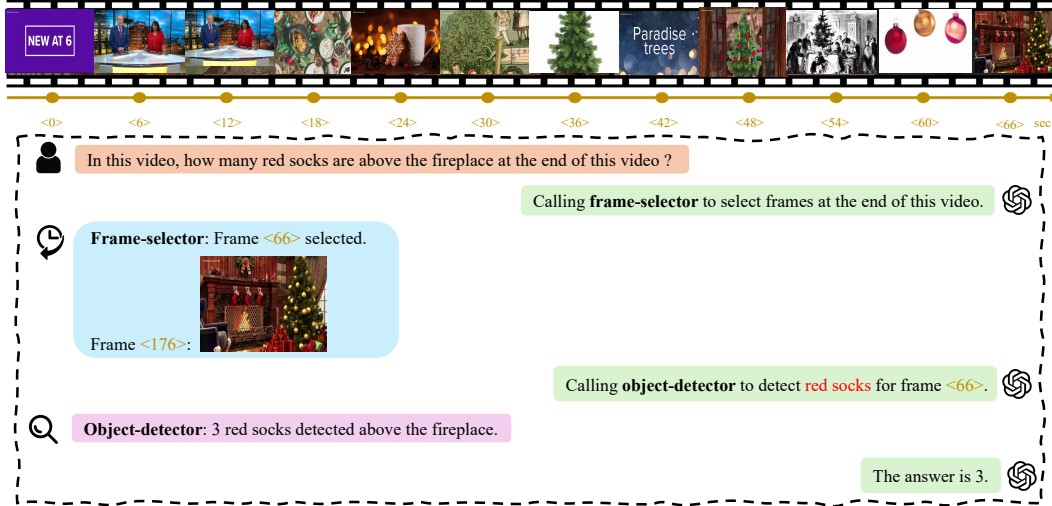

Figure 4: A case of counting problem from VideoMME, mainly solved by object detector.

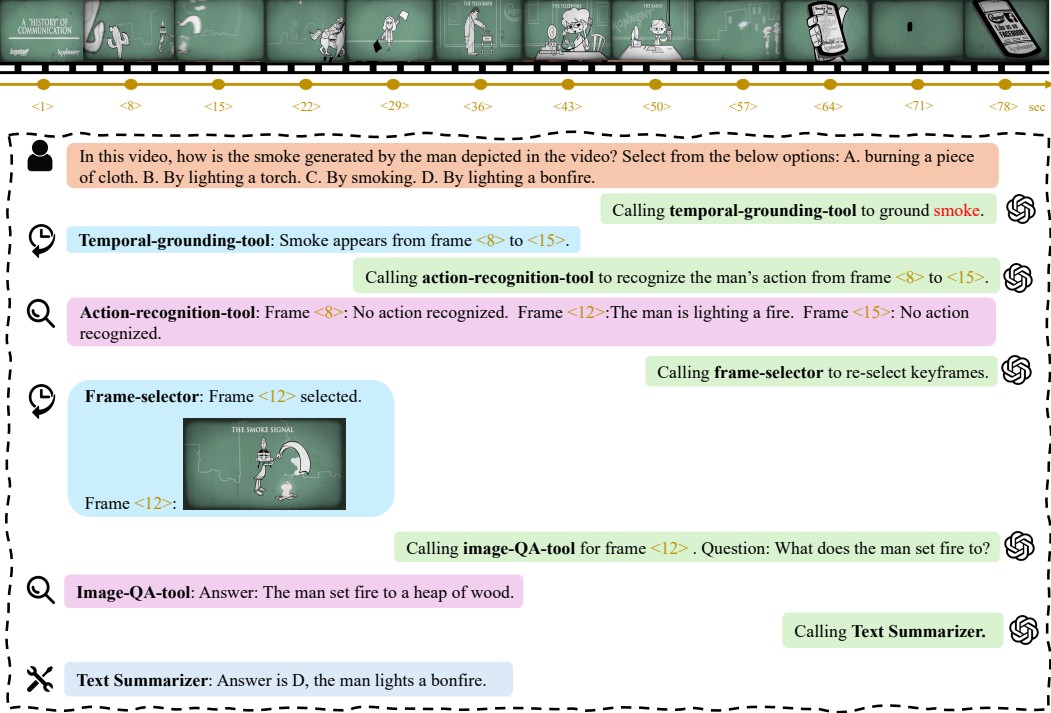

Figure 5: A case of action recognition problem from VideoMME.

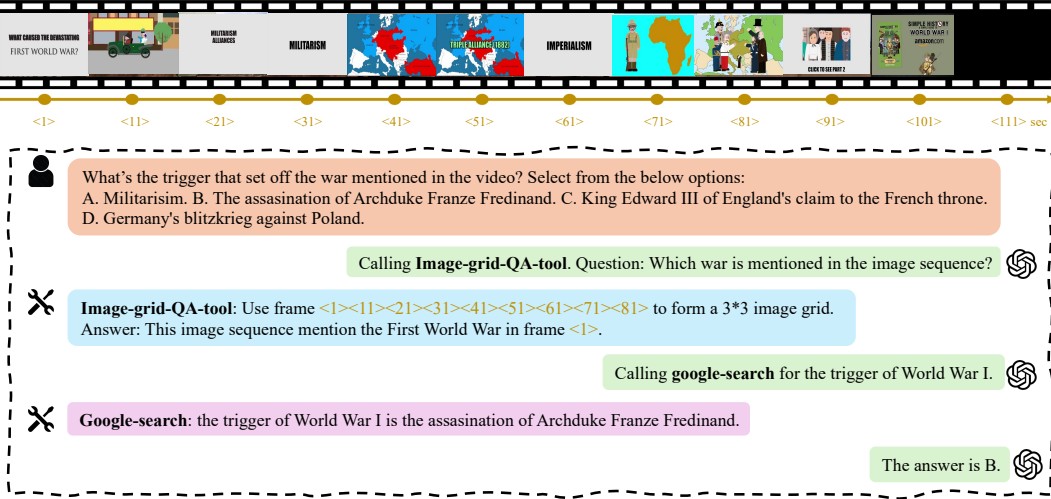

Figure 6: A case of action reasoning problem from VideoMME.

