# OpenReview forum: "Tool-Augmented Spatiotemporal Reasoning for Streamlining Video Question Answering Task"
_NeurIPS.cc/2025/Conference — NeurIPS 2025 poster_

### Official Review · Reviewer_Fvz6 · 2025-06-26

**Clarity:** 2
**Significance:** 3
**Originality:** 3
**Rating:** 4
**Confidence:** 3

**Summary:**

To better control the tool invocation sequence and avoid toolchain shortcut issues, the authors propose a Spatiotemporal Reasoning Framework (STAR) that strategically schedules temporal and spatial tools, thereby progressively localizing the key area in the video. The STAR framework enhances GPT-4o using lightweight tools, achieving an 8.2% gain on VideoMME and 4.6% on LongVideoBench.

**Questions:**

See weakness.

**Ethical Concerns:**

["NO or VERY MINOR ethics concerns only"]

**Final Justification:**

Thanks for the rebuttal. I will maintain the original score.

**Limitations:**

See weakness.

**Quality:**

3

**Strengths And Weaknesses:**

W1. Baseline performance concern: The reported VideoMME performance of Qwen2.5VL is 71.9. However, the result marked by the authors is 61.8, which shows a discrepancy of nearly 10 points. Could the authors please explain the reason for this difference, provide the exact prompt used, or alternatively test their method on the open-sourced Qwen or InternVideo models? Doing so would make the results more convincing.

W2. In Figure 3, the categorization of the Video Summarizer differs from that in the supplementary materials. Could the authors clarify the reason for this inconsistency?

W3. Regarding efficiency, the authors are encouraged to provide more detailed information about the missing Runtime values in the table, as well as clarify the rather vague description of "<30s".

S: The use of tools is reasonable, with a well-defined and detailed categorization of tool types. Moreover, the performance improvement appears to be significant.

---

> ### Author Rebuttal · Authors · 2025-07-27
>
> We sincerely thank you for your selfless efforts and constructive feedback, which greatly helps us improve the quality of this work. We are especially grateful for the recognition of our categorization of tool types, as well as the performance improvement. Below, we provide point-by-point responses for your concerns.
>
> ## W1
>
> > Discrepancy in reported Video-MME score and lack of prompt transparency
>
> In Table 1, the score of 71.9 corresponds to GPT-4o evaluated with a sampling rate of 1 fps, with up to 384 frames sampled per video. The score of 61.8 corresponds to GPT-4o evaluated with a fixed sampling of 32 frames per video.
>
> The accuracy of video question answering is closely tied to the number of sampled frames—higher sampling rates generally yield better performance. Therefore, fair comparisons should be conducted under comparable sampling conditions. Since our method samples an average of 30.2 frames per video, we compare against GPT-4o’s 32-frame setting (61.8) and mark 61.8 accordingly.
>
> The sources and evaluation method of the results reported in Table 1 are as follows, all under the *without subtitles* setting:
>
> | Model             | Sample Rate / Num of sampled Frames | Accuracy (%) | Source or Eval Method                                        |
> | ----------------- | ----------------------------------- | -------- | ------------------------------------------------------------ |
> | GPT-4o            | 1 fps / 384                         | 71.9     | Video-MME official leaderboard                               |
> | GPT-4o            | 32                                  | 61.8     | self-eval by Video-MME recommended way (using `lmms-eval` Github repository) |
> | GPT-4o + T*       | 32                                  | 64.1     | T* paper Table 4                                             |
> | GPT-4v            | 10                                  | 60.0     | Video-MME official leaderboard                               |
> | Gemini 1.5 Pro    | 0.5 / 1 fps                         | 75.0     | Video-MME official leaderboard                               |
> | Claude 3.5 Sonnet | 20                                  | 60.9     | Video-MME official leaderboard                               |
> | Qwen2-VL-72B      | 2 fps / 768                         | 71.2     | Video-MME official leaderboard                               |
> | Qwen2-VL-7B       | -                                   | 63.3     | InternVL3 tech report Table 8                                |
> | Qwen2.5-VL-7B     | -                                   | 65.1     | InternVL3 tech report Table 8                                |
> | InternVL2.5-8B    | 64                                  | 64.2     | InternVL3 tech report Table 8                                |
> | InternVL3-8B      | 64                                  | 66.3     | InternVL3 tech report Table 8                                |
> | VideoLLaMA3       | 180                                 | 66.2     | Video-MME official leaderboard                               |
> | mPLUG-Owl3        | 128                                 | 59.3     | Video-MME official leaderboard                               |
>
> All time measurements are conducted using either official APIs or open-source models, following the Video-MME recommended protocol (via the `lmms-eval` GitHub repository). Each measurement is repeated multiple times (at least 3) on 8 NVIDIA GeForce RTX 4090 GPUs.
>
> ## W2
>
> > Categorization inconsistency of Video Summarizer between the main paper and the supplementary materials
>
> **There is a classification error in Figure 3: Video Summarizer should be categorized under *General Tool* rather than the *Both* category. The information provided in the appendix is correct.**
>
> *General Tools* are only used at the end of the toolchain, when neither temporal nor spatial tools can resolve the question. In contrast, *Both* tools can function as either temporal or spatial tools within the toolchain. The rationale for classifying **Video Summarizer** as a *General Tool* is that it is implemented using a 3B MLLM and is intended for use only as a last resort. In typical cases, we prioritize lightweight tools to improve computational efficiency. We will correct this classification in the later version.
>
> ## W3
>
> > Lack of clarity in runtime reporting and missing efficiency details
>
> ### 1. **About the missing Runtime values**
>
>   There is a strong positive correlation between runtime and the number of sampled frames. **The runtime value is missing for that particular data point in the table because it was taken from another paper that did not report the number of sampled frames, making it impossible to accurately estimate the runtime.**
>
>   This includes two data points in Table 1—**Qwen2-VL-7B** and **Qwen2.5-VL-7B**—which are sourced from Table 8 of the InternVL3 technical report, as well as two data points in Table 4—**ViperGPT** and **VideoChat**—which are sourced from Table 2 of the DoraemonGPT paper.
>
>   We thank the reviewer for pointing out this limitation. In the later version, we plan to replace these externally reported results with those obtained from our own reproduction, thereby eliminating the missing runtime values.
>
> ### 2. **About the vague description of "<30s"**
>
>   **These runtime measurements are inherently difficult to obtain accurately, as they are sensitive to various environmental factors**: (1) The speed of API is affected by network conditions; (2) Even with the same GPU model, different compute clusters may exhibit variations in video processing speed due to factors such as CPU memory, system age, and overall hardware configuration.
>
>   Moreover, runtime is highly dependent on the number of frames, making fine-grained time comparisons meaningless when the number of frames differs slightly.
>
>   Therefore, we run each baseline experiment multiple times to obtain a rough performance range, e.g., <30s. Regardless of the interference factors, the timing of these methods generally falls within this range. For our own method, STAR, we conduct more extensive testing (around 10 runs) under different environments to obtain a more accurate average reported in the table, e.g. 15.8s.
>
>   **Our intention is to determine approximate performance ranges**—such as *<30s, 30s–60s, or >10min*—in scenarios where precise runtime measurement is challenging. Models falling within the same range can be considered to have comparable computational efficiency, as small differences within a range are unlikely to impact practical usage.
>
>   **Our intention is also not to underrepresent baseline performance**, but rather to show:
>   (1) STAR (averaging 15.8 seconds per video with 30.2 sampled frames) operates in the same time scale as models like GPT-4o (0–30 seconds for 32 sampled frames).
>   (2) STAR achieves comparable accuracy to GPT-4o (>10 minutes for 384 sample frames) but completes processing much faster.
>
>   **In future revisions, we will conduct additional experiments across diverse environments to obtain more precise runtime measurements wherever possible.**

---

### Official Review · Reviewer_ah2Q · 2025-07-03

**Clarity:** 3
**Significance:** 2
**Originality:** 3
**Rating:** 5
**Confidence:** 3

**Summary:**

When MLLMs do VideoQA, it is hard for the model to comprehend spatial relationships and temporal evolution. The author equips an extensible video toolkit that contains three types of tools: spatial tools, temporal tools, and general tools, on MLLMs to help MLLMs perform better on spatiotemporal reasoning. In addition, the author proposes the Spatiotemporal Reasoning Framework(STAR), which could arrange the use of “spatial tools” and “temporal tools” and controls the tool invocation sequence, called spatiotemporal-interleaved toolchain and uses tools with spatial localization capabilities, to compensate for the shortcomings of MLLMs in spatial localization. Moreover, STAR could also help avoid toolchain shortcut issues and find the key area in the video. This method performs well in multiple video question answering tasks(VideoQA).

**Questions:**

1. The paper mentioned that there are 30 tools in the toolkit, but only 22 are described in the appendix. Do the remaining tools appear to be omitted or described elsewhere?

2. The method seems to be portable, but it seems tha the authors only tested it on GPT-4o. I'm interested to know how well it performs with other backbone models, especially open-source ones like Qwen or Llama.

3. What if the model chooses the wrong time at the beginning? Can the model correct the error?

**Ethical Concerns:**

["NO or VERY MINOR ethics concerns only"]

**Final Justification:**

Thanks for the responses. The responses have successfully addressed my concerns. Although the paper is not particularly interesting, it is solid and well-written, so I will raise my score to 5.

**Limitations:**

Yes

**Quality:**

2

**Strengths And Weaknesses:**

Strength:
- This paper is well written, with clear figures that help readers understand.
- STAR is very portable, as the author designed the STAR mini that could run on a MAC.
- The author compared STAR to various other methods, including no constraints, prompting, in-context learning, and spatiotemporal disentanglement, showing that STAR has the best accuracy and frame efficiency.
- The authors tested STAR on various datasets, including VideoMME, NExT-QA, LongVideoBench, and EgoSchema.

Weakness
- All the tools are plug-and-play, which could lead to the model not being able to study from previous experiments.
- For VideoMME, STAR gets a 70% accuracy; however, according to Zero-Shot Video Question Answer on Video-MME (https://paperswithcode.com/sota/zero-shot-video-question-answer-on-video-mme-1), there is already a model that gets an 81.3% accuracy, which is higher than the results claimed in the paper.

---

> ### Author Rebuttal · Authors · 2025-07-26
>
> Thank you for your thoughtful and encouraging feedback. We are especially grateful for your recognition of STAR’s portability, superior performance across multiple methods, and strong results on diverse benchmarks with clear visualizations. The followings are points-to-points respones to your concerns.
>
> ## Weaknesses
>
> ### W1
>
> > Model not being able to learn from previous experiments
>
> We thank the reviewer for pointing out this limitation. We acknowledge that **tool-integrated post-training** is an important direction and will leave it as **future work** to be explored in follow-up studies.
>
>    This paper focuses on a **training-free, plug-and-play** paradigm, where tools are dynamically selected and invoked without additional fine-tuning. In practice, we have found that current LLM such as GPT-4o already exhibit strong capabilities in instruction-following and zero-shot tool use. As a result, **training-free approaches** can achieve competitive performance without the need for additional fine-tuning. We believe this paradigm offers a complementary alternative to **tool-integrated post-training**, with each approach having its own trade-offs.
>
> ### W2
>
> > Why some results on Video-MME is higher than 80%? Is STAR's 70% accuracy a good performance? Details about Video-MME evaluation
>
> This is primarily due to two reasons:
>
>    (1) **Video-MME provides two evaluation modes: *with subtitles* and *without subtitles***. Models generally perform better in the *with subtitles* setting. For example, on the official Video-MME leaderboard, Gemini 1.5 Pro achieves 81.3% accuracy with subtitles, compared to 75.0% without subtitles. Our work currently focuses exclusively on pure video understanding, without leveraging subtitle information. **Therefore, in Table 1, we report results under the *without subtitles* setting**. In contrast, many entries on the PapersWithCode leaderboard either report *with subtitles* results or do not specify the evaluation setting, making direct comparison less reliable.
>
>    (2) **The accuracy of video question answering is closely related to the number of sampled frames**—higher sampling rates generally lead to better performance. Therefore, fair comparisons require evaluations under the same sampling rate. Some results reported on PapersWithCode may be obtained under higher sampling rates.
>
> To ensure fairness, we also report our performance under increased sampling rates as follows. Due to API budget and time constraints, we randomly sampled 1,000 examples from the Video-MME test set for this evaluation. The following results demonstrate that the STAR framework continues to improve performance as the number of sampled frames increases and reach 77% performance under the *without subtitles, 1fps sampling, up to 384 frames* condition.
>
> | Model  | LLM Planner | Average Frames | Short (%) | Medium (%) | Long (%) | All (%)           |
> | ------ | ----------- | -------------- | --------- | ---------- | -------- | ----------------- |
> | GPT-4o | -           | 32             | 68.6      | 60.6       | 56.0     | 61.5              |
> | GPT-4o |             | 100            | 72.8      | 63.2       | 59.5     | 64.9              |
> | GPT-4o | -           | 1 fps / 384    | 79.2      | 70.4       | 66.5     | 71.8              |
> | STAR   | GPT-4o      | 31.3           | 78.2      | 68.7       | 62.7     | 69.6 (8.1% ↑)     |
> | STAR   | GPT-4o      | 100.2          | 80.1      | 71.0       | 66.4     | 72.4 (7.5% ↑)     |
> | STAR   | GPT-4o      | 0.98 fps / 384 | **85.3**  | **78.0**   | **68.5** | **77.0** (5.2% ↑) |
>
> ## Questions
>
> ### Q1
>
> > Tool number discrenpency between the main paper and appendix
>
> The number *22* is the number of tool types, while *30* is the number of tool implementations. A tool may have multiple implementations to accommodate different computational resources or deployment environments. For example, the **Frame Selector** has 3 variants:
>
>    + **Vanilla version**: The frame selector is an LLM that takes as input the IDs and textual descriptions of all visible frames (i.e., the *Visible Frame Dictionary* described in the paper), such as captions generated by an image captioner. It outputs the selected frame IDs. The prompt is directly instructing the LLM to select relevant frames.
>
>    + **AKeyS-inspired version**: This variant shares the same input and output format as the Vanilla version but employs specially designed prompts to encourage the LLM to consider task goals and scene segmentation.
>
>    + **T\*-inspired version**: In this case, the frame selector is an MLLM, and its input additionally includes the actual images of visible frames. These images are further arranged into an image grid to facilitate selection. The output remains the same.
>
>    Similarly, the **Object Detector** has both YOLO- and GroundingDINO-based versions tailored for low- and high-resource settings, respectively. The **Image Captioner** and **Image QA Tool** have different implementations based on BLIP2 (\~1B parameters), LLaVA (\~7B), and GPT-4o.
>
>    The number *30* mentioned in the paper refers to the total number of tool implementations. **We acknowledge that this may cause confusion and will revise in later version to clarify this distinction and eliminate ambiguity.**
>
> ### Q2
>
> > Performance with other backbone models
>
> We also evaluated our framework with other (M)LLMs, and the results are as follows. Due to API budget and time constraints, we randomly sampled 1,000 examples from the Video-MME test set for this evaluation.
>
>    | Model          | LLM Planner    | Average Frames | Short (%)    | Medium (%)   | Long (%)     | All (%)            |
>    | -------------- | -------------- | -------------- | -------- | -------- | -------- | ------------------ |
>    | GPT-4o         | -              | 32             | 68.6     | 60.6     | 56.0     | 61.5               |
>    | Gemini-2.5-pro | -              | 32             | 77.6     | 62.6     | 57.1     | 65.4               |
>    | Qwen2.5-VL-72B | -              | 32             | 68.9     | 58.8     | 55.4     | 60.8               |
>    | STAR           | GPT-4o         | 31.3           | 78.2     | 68.7     | 62.7     | 69.6 ( 8.1 ↑ ) |
>    | STAR           | Gemini-2.5-pro | **31.0**       | **79.8** | **70.7** | **69.1** | **72.9** ( 7.5 ↑ ) |
>    | STAR           | Qwen2.5-VL-72B | 31.5           | 77.9     | 66.1     | 62.4     | 68.5 ( 7.7 ↑ )     |
>    | STAR           | DeepSeek-R1    | 31.2           | 77.2     | 67.2     | 63.0     | 68.9               |
>
>    The results show that both closed-source models (e.g., GPT-4o, Gemini 2.5 Pro) and open-source models (e.g., Qwen2.5-VL-72B, DeepSeek-R1) achieve strong performance. Compared to their tool-free counterparts, these models demonstrate a **7%–8% improvement in accuracy**, highlighting the effectiveness of our work across model types.
>
> ### Q3
>
> > Error correction after initial temporal misalignment
>
> It is normal for our STAR framework to initially focus on an incorrect video segment. However, our STAR framework is equipped with backtracking ability that enables correction and prevents cascading errors. When the LLM Planner first invokes the frame selector and the Visible Frame Dictionary is still empty, the selector performs a coarse, uniform sampling over the entire video, dividing it into several segments. It then chooses to explore one segment in depth. **If, after comparing the question with the retrieved information, it turns out that the current segment is not relevant, the frame selector is able to shift its focus to other segments.** The exploration process terminates only when the LLM Planner has gathered sufficient information to answer the question accurately.
>
> For example, in a case study from the Video-MME test set (`question_id=301-2`), the question is: *“Why can Cusco and Tenochtitlan be easily found while Amazonian settlements cannot?”*. The associated video is a YouTube video titled *“How the ‘lost cities’ of the Amazon were finally found”*. Initially, STAR selected frames from the segment between 1:20 and 1:40, which only mentioned Cusco and Tenochtitlan, but did not address why Amazonian settlements were hard to find. Detecting the lack of relevant information, STAR then shifted to another segment and ultimately identified the correct time span (6:10–6:20), which states: *“Because Amazonian settlements were built with wood and earth.”*

---

> ### Comment · Reviewer_ah2Q · 2025-08-08
> **Thanks for the rebuttal**
>
> Thanks for the responses. The responses have successfully addressed my concerns. Although the paper is not particularly interesting, it is solid and well-written, so I will raise my score to 5.

---

> ### Author Response · Authors · 2025-08-08
>
> We sincerely thank you for your thoughtful comments and careful consideration. We truly appreciate your decision to raise the score. Your constructive feedback has been very helpful in improving our work.

---

### Official Review · Reviewer_JAXG · 2025-07-03

**Clarity:** 3
**Significance:** 3
**Originality:** 3
**Rating:** 4
**Confidence:** 4

**Summary:**

This paper introduces a plug-and-play Spatiotemporal Reasoning Framework (STAR), which addresses a critical limitation in existing Video Toolkit-augmented MLLMs: their inability to effectively integrate spatiotemporal information for complex, reasoning-intensive VideoQA. The proposed framework incorporates a novel tool scheduling strategy that optimizes the fusion of complementary spatiotemporal cues, enabling more robust understanding of 3D Regions of Interest (3D RoIs). Through extensive experiments, STAR demonstrates significant improvements over existing approaches, achieving an 8.2% gain on VideoMME [12] and a 4.6% gain on LongVideoBench [62] when augmenting GPT-4o with its lightweight toolkit.
The paper is well-structured, with clear and intuitive figures that enhance readability, reflecting the authors’ experience in rigorous scientific writing. The experimental validation is thorough, encompassing multiple video understanding benchmarks (eg.VideoMME) and diverse state-of-the-art models (eg. Open-source Video-LLMs), which collectively substantiate the framework’s effectiveness and generalizability. These results validate STAR’s capability to bridge the gap between spatial and temporal reasoning, offering a versatile and extensible solution for advanced video understanding.

**Questions:**

1. Experimental Comparisons (Time & Frame Processing)

1)Tool Call Time Reduction: The results show that using tools is faster than raw GPT-4o processing. How is this possible?
Is the tool execution time negligible (e.g., due to parallelization or caching)?
Were any specific acceleration techniques applied (e.g., pre-filtering frames, early termination)?

2)Reduction in Processed Frames: The decrease in processed frames is attributed to STAR’s selective reasoning, but:
Could this also be influenced by additional textual information (e.g., metadata or captions) reducing the need for full-frame analysis?
If so, is the frame reduction truly due to spatiotemporal reasoning, or partly due to auxiliary inputs?

2. Tool Quality & Harmony Assurance (Abstract vs. Implementation)

The abstract claims "ensuring tool quality and harmony", but the paper lacks details on how this is achieved.
Is this solely reliant on the Planner + carefully crafted prompts, or are there additional mechanisms (e.g., tool validation, fallback strategies)? If prompt engineering is the key, how is robustness guaranteed across diverse queries?
Are there quantitative metrics (e.g., success rate, error recovery cases) to support this claim?

**Ethical Concerns:**

["NO or VERY MINOR ethics concerns only"]

**Final Justification:**

They have addressed my concern. I will maintain my score.

**Limitations:**

This work appears to have minimal potential for negative societal impact.

**Paper Formatting Concerns:**

No major formatting issues are found in this paper.

**Quality:**

3

**Strengths And Weaknesses:**

Strengths：

1. The paper introduces a novel tool scheduling strategy that effectively optimizes the fusion of complementary spatiotemporal cues via interleaved spatial and temporal tool utilization.
2. The proposed approach successfully mitigates the Toolchain Shortcut issue and enables more robust 3D Regions of Interest (3D RoIs) understanding, as validated by extensive benchmarking.

Weaknesses：

Typo：

1. The text in Lines 40-46 appears somewhat abrupt. Please reorganize the logical flow for better coherence.
2. Regarding "(3rd place) Toolchain Shortcut" in Figure 1: Could you clarify what this specifically refers to?
3. For the comparison with DoraemonGPT in Line 95: Please provide more detailed explanations to substantiate the claims of superior reliability and efficiency.
4. In Line 106: Should "In the temporal dimension" perhaps be "In the spatiotemporal dimension" instead? This would better reflect the comprehensive analysis being presented.

Core Concerns and Questions:

1. Toolkit Design:

Does the tool include a custom-designed toolkit (e.g., integrating multiple existing tools/models to form a specialized solution for a specific problem class), rather than relying solely on individual off-the-shelf models/tools?

2. Frame Index Adjustment (Line 166): The statement “by adding or removing frame indices as needed” raises questions: When would frames be added? (Line 55 also mentions expansion—under what conditions?) How is it determined which frames to add and how many? (Removal is straightforward via index deletion, but addition logic is unclear.)

3. Toolchain Shortcut Definition (Line 149): The definition “We define Toolchain Shortcut as a phenomenon…” prompts a concern:
Could cases exist where a general tool alone suffices, but this method unnecessarily complicates the process? Is there a mechanism to distinguish between simple (single-step solvable) and complex (step-by-step toolchain required) problems?
4. Spatiotemporal Interleaving (Line 178 & Fig. 2): The example in Fig. 2 shows temporal → spatial tool order. However，if a spatial tool (e.g., frame captioning) is called first, followed by temporal localization, would the system re-invoke spatial tools afterward? Is the temporal-first preference inherent to most problem types, or is the order flexible?
5. Relation to Prior Work (Lines 184–189): How does the proposed STAR reasoning compare to prior methods that decompose queries stepwise to identify required tools/information? Are there key differences or improvements over existing query decomposition + tool retrieval approaches?
6. Experiments (Toolkit Generalizability): The claim “we equip (M)LLMs with a carefully designed and comprehensive video toolkit” suggests broad applicability: Beyond GPT-4o, were other MLLMs tested with this toolkit? Could additional experiments (e.g., applying the toolkit to other MLLMs like LLaMA or Gemini) strengthen the generalizability argument?

---

> ### Author Rebuttal · Authors · 2025-07-26
>
> We sincerely thank you for your constructive feedback. We are especially grateful for your recognition. Below, we provide point-by-point responses for your concerns:
>
> ## Weaknesses
>
> ### Typos
>
> 1. Thank you for your feedback. We will revise this paragraph to ensure greater logical coherence.
> 2. The "(3rd place) Toolchain Shortcut" means that the *Toolchain Shortcut* method performs worst among the three approaches in figure 1. For a more detailed description and corresponding results, please refer to Section 4.2 and Table 5 in the main paper.
> 3. DoraemonGPT use tools to processes all frames of a video to obtain information, which are then stored in a SQLite database. An LLM subsequently translates natural language queries into SQL statements to retrieve answers from the database. However, since DoraemonGPT operates on all video frames—unlike the STAR framework, which temporally narrows the search scope—**it is less efficient**. Moreover, due to limitations in text-to-SQL accuracy, **the generated SQL queries in DoraemonGPT sometimes fail to execute, compromising reliability**. In contrast, STAR avoids such failure-prone intermediate query steps.
> 4. The original intent of line 106 is to emphasize that the Video Toolkit should be capable of handling both individual frames and video sub-clips—this is considerations along the **temporal dimension**. Correspondingly, along the **spatial dimension**, Video Toolkit should support operations on both full images and localized image patches. In the revised version, we will explicitly discuss both the temporal and spatial dimensions to reflect comprehensive analysis.
>
> ### Core Concerns and Questions
>
> > 1. About custom-designed tools
>
> **Our current Video Toolkit does not yet include custom-designed tools. However, when develop the Video Toolkit, we consider various classic problem classes** such as counting, action reasoning, text reasoning problems, and so on. The Video Toolkit support the full solution pipeline needed to address these representative problems. In future work, we plan to develop custom-designed toolkits and solution pipelines, and integrate them with the LLM Planner’s autonomous tool-calling mechanism.
>
> > 2. When and How to add frames?
>
> The following temporal tools, once invoked and executed, each apply their own selection logic to expand the set of visible frames.
>
>    + **Frame Selector** is an (M)LLM-based tool that, guided by single-turn or multi-turn prompting, analyzes the current Visible Frame Dictionary and identifies additional unseen frames that are likely to be informative for answering the question. These selected frames are then incorporated into the visible set to support further reasoning.
>
>    + **Temporal Grounding Tool** and **Action Localization Tool**: When invoked by the LLM Planner, the tool takes a textual description as input and grounds it to a corresponding temporal segment in the video. Frames from this segment are then sampled at 1 fps and added to the set of visible frames.
>
> > 3. Toolchain shortcut definition
>
> More rigorously, *toolchain shortcut* refers to cases where **the toolchain invoked by the LLM Planner is shorter than the ground-truth toolchain.** The *ground-truth toolchain* represents the most computation-efficient sequence of tool calls required to solve the task. Cases where “a general tool alone suffices” should not be categorized as toolchain shortcuts. In the upcoming revision of this paper, we will manually annotate the ground-truth toolchains for the dataset and compare them with those generated by the LLM Planner, thereby enabling a more precise characterization of the toolchain shortcut phenomenon and examining whether there are cases that unnecessarily complicate the process.
>
> > 4. Spatialtemporal interleaving detail
>
> The older is flexible as we mention in line 172. In other words, the first tool in the chain can be either a temporal or a spatial tool, but the subsequent tools are required to alternate between temporal and spatial tools.
>
> > 5. Comparison with "query decomposition" works
>
> There are 2 main differences:
>
>    + Prior works [1, 2] about query decomposition is better suited for **complex multi-step reasoning tasks where each step requires a different tool**. For example, Chain-of-Action [1] focuses on retrieving real-time information from heterogeneous sources. In contrast, our approach focuses on **decomposing the process of locating key regions into a step-by-step procedure**, which is better suited for real-world long-form video QA tasks depending on accurate localization of key frames and regions.
>    + STAR determines which tool to invoke and what action to take step by step during the exploration of the video environment, rather than predefining the entire reasoning trajectory based solely on the query. This design makes the framework more flexible and dynamic.
>
>    In future work, we will further investigate this comparison and aim to validate it experimentally.
>
>    References:
>
>    [1] Chain-of-action: Faithful and multimodal question answering through large language models.
>
>    [2] ToolACE-DEV: Self-Improving Tool Learning via Decomposition and EVolution.
>
> > 6. Performance with other MLLMs
>
> Due to API budget and time constraints, we randomly sampled 1,000 examples from the Video-MME test set for this evaluation. The results show that, compared to their tool-free counterparts, **these models demonstrate a 7%–8% improvement in accuracy, highlighting the effectiveness across model types.**
>
>
> | Model          | LLM Planner    | Average Frames | Short (%)   | Medium (%)   | Long (%)    | All (%)               |
> | -------------- | -------------- | -------------- | -------- | -------- | -------- | ------------------ |
> | GPT-4o         | -              | 32             | 68.6     | 60.6     | 56.0     | 61.5               |
> | Gemini-2.5-pro | -              | 32             | 77.6     | 62.6     | 57.1     | 65.4               |
> | Qwen2.5-VL-72B | -              | 32             | 68.9     | 58.8     | 55.4     | 60.8               |
> | STAR           | GPT-4o         | 31.3           | 78.2     | 68.7     | 62.7     | 69.6 ( 8.1 ↑ ) |
> | STAR           | Gemini-2.5-pro | **31.0**       | **79.8** | **70.7** | **69.1** | **72.9** ( 7.5 ↑ ) |
> | STAR           | Qwen2.5-VL-72B | 31.5           | 77.9     | 66.1     | 62.4     | 68.5 ( 7.7 ↑ )     |
> | STAR           | DeepSeek-R1    | 31.2           | 77.2     | 67.2     | 63.0     | 68.9               |
>
> ## Questions
>
> ### Experimental Comparisons
>
> > 1. Why STAR faster than raw GPT-4o?
>
> The main reason is that in our work, the video is not directly fed into GPT-4o. Instead, GPT-4o—serving as the LLM Planner—analyzes the question along with intermediate textual results stored in the Visible Frame Dictionary to decide which tool to invoke next. The LLM Planner does not process video directly. **Therefore, In our work avoids processing a large number of image tokens.**
>
> > 2. What cause the reduction in processed frames?
>
> We believe the key lies in **selective reasoning**.
>
> - **Regarding metadata**: Our framework does not introduce any additional metadata—the input remains a single video and a question.
> - **Regarding captions**: If the LLM Planner invokes an tool (e.g., an image captioner) on a particular frame, that frame is counted as a *sampled/visible/processed* frame. Therefore, captions are not considered auxiliary inputs but rather *intermediate tool results*.
>
> As such, the system **does not rely on auxiliary inputs**. However, it can be interpreted as using tools to extract *different kinds of information* from selected frames. This spatiotemporal reasoning approach enables the QA system to selectively fuse and utilize diverse information sources, allowing it to reduce the number of processed frames while maintaining high accuracy.
>
> ### Tool Quality & Harmony Assurance
>
> > How to improve tool quality and harmony? How to measure?
>
> We use the following methods to improve **tool quality** and ensure robust tool execution:
>
> - **Structured tool invocation**: We leverage LLM's (e.g., GPT-4o)  capability for structured output by requiring the LLM Planner to generate tool parameters in JSON format. These parameters are then parsed to ensure syntactic correctness.
> - **Time range check**: We ensures that temporal manipulations such as frame sampling or trimming are accurately bounded within the valid time range of the video, preventing out-of-bounds errors.
> - **Resilient workflow design**: The workflow is designed to be fault-tolerant. A single failed or ineffective tool invocation does not halt the process. For instance, if a spatial tool like an object detector fails to detect the target object, the planner may subsequently invoke a temporal tool to adjust frame sampling and continue exploration.
> - **Fallback General Tools**: We include general-purpose tools, such as the Video QA Tool and Video Summarizer based on video-language models, as a fallback mechanism. These tools serve as a safety net when specialized tools fail to yield sufficient information.
>
> In terms of **metrics**, we adopt the following approaches to evaluate tool execution and usage quality:
>
> - **Tool Success Rate**: We measure the proportion of successful tool executions—defined as invocations where input parameters are correct and the tool returns valid outputs. On the Video-MME test set, the success rate is **97.2%**, and on the LongVideoBench test set, it is **96.5%**, indicating that tools are generally executed reliably.
> - **Tool Usage Distribution**: To assess the **balance and harmony** of tool usage, we analyze the distribution of tool calls across different tools and tool categories. This helps us evaluate whether certain tools are disproportionately relied upon, and whether the LLM Planner effectively utilizes the full range of available tools. The results of this metric can be found in Figure 3 of the main paper, as well as in our rebuttal to Reviewer QhNP, Weakness 3.

---

> > ### Comment · Reviewer_JAXG · 2025-08-07
> >
> > The authors have largely addressed my concerns. After carefully considering their rebuttal and the contributions of the paper, I have decided to maintain the original score.

---

### Official Review · Reviewer_QhNP · 2025-07-05

**Clarity:** 3
**Significance:** 3
**Originality:** 3
**Rating:** 5
**Confidence:** 4

**Summary:**

This paper presents STAR, a tool-augmented spatiotemporal reasoning framework for video understanding. The authors construct a diverse set of spatial tools, temporal tools and general tools, then use the STAR algorithm to coordinate all tools for video understanding with an LLM as the controller. To prevent toolchain shortcut, the authors also design a spatiotemporal interleave tool calling strategy. Extensive experiments on Video-MME and LongVideoBench show strong performance of the proposed method.

**Questions:**

Please see the ‘Weaknesses’ section. Overall, the reviewer thinks that the method is intuitive, interesting, simple and works well. However, it would be good if the authors could clarify the implementation details and show that the proposed method also performs well with more frames. The reviewer will consider raising the score if the above weaknesses are solved.

**Ethical Concerns:**

["NO or VERY MINOR ethics concerns only"]

**Final Justification:**

Thanks for the rebuttal. My main concerns are well addressed, especially for the missing implementation details and the performance of the proposed method with more frames. The method is interesting, well-motivated and works well on many datasets. Therefore, I will raise my score to 5. I hope that the authors add the missing implementation details and the new experimental results to the final paper.

**Limitations:**

It would be good to provide some failure case analysis to help people understand more about the limitations of the proposed method.

**Quality:**

3

**Strengths And Weaknesses:**

Strengths:

- Overall, the flow of paper writing is clear and easy-to-follow.
- The proposed method is simple, intuitive, training-free, yet works well. Experiments on VideoMME and LongVideoBench show that the proposed method outperforms multiple baselines with a smaller runtime.
- The qualitative examples are clear, intuitive, and can help people understand the method.

Weaknesses:

- Some important implementation details are not clearly presented.
    - What is the controller LLM?
    - How many frames are sampled initially?
    - In Table 1, how is the runtime calculated? What GPUs did the authors use? Is the comparison with other methods apple-to-apple? Why are some runtimes very vague? For example, the runtime of GPT-4o is presented as <30s, but the runtime of the proposed method which uses GPT-4o is clearly presented as 15.8s.  Without standardized evaluation conditions and transparent reporting, it is challenging to assess the actual efficiency advantage of the proposed method over the baselines.
    - The introduction to the frame selector tool is not clear. Is the frame selector tool an LLM taking captions as input and output frame IDs, or video trimming, or frame selection methods as AKeyS and T*?
- It is good that the proposed method outperforms GPT-4o, with both methods taking ~30 frames as input. However, the toolset of the proposed method includes GPT-4o itself. This raises the question: how does the proposed method perform when provided with more frames, such as in GPT-4o’s 1 fps / 384 setting? It would significantly strengthen the paper if the authors could show that their method continues to outperform GPT-4o even when both are using the same number of frames (e.g., 1 fps / 384).
- As the authors mention, one of the motivation is to solve the ‘Unbalanced number and diversity of tools’ issue. However, it is not clear how exactly the authors solve this issue. From the reviewer’s understanding, the spatiotemporal interleave tool calling strategy can balance the spatial tools and temporal tools, but what about the tools within the same categories?
- It would be good to provide some failure case analysis to help people understand more about the limitations of the proposed method.

---

> ### Author Rebuttal · Authors · 2025-07-25
>
> We sincerely thank you for your constructive feedback, which greatly helps us improve the quality of this work. Below, we provide point-by-point responses for your concerns:
>
> ##  Weakness 1
>
> > Some missing implementation details
>
> + For STAR framework，we use **gpt-4o-2024-08-06** as the controller LLM. For STAR-mini, we use **gpt-3.5-turbo-0125** as the controller LLM to ensure a fair comparison with baselines.
>
> + For videos longer than 16 seconds, we extract 16 initial frames uniformly; for videos with a duration of 16 seconds or less, initial frames are extracted at a rate of 1 fps.
>
> + For proprietary models such as GPT-4o, we use their official APIs. For open-source models such as Qwen, we benchmark models up to 8B on 8 NVIDIA RTX 4090 GPUs with vLLM, and benchmark 72B models on 8 NVIDIA H100 GPUs with vLLM. For both prior work such as T* and our proposed STAR framework, experiments are conducted on 8 NVIDIA RTX 4090 GPUs using the same official APIs, ensuring a relatively fair comparison.
>
>   **Why are some runtimes very vague?** This is primarily due to three reasons:
>   1. The speed of API is affected by network conditions;
>   2. Even with the same GPU, different compute clusters may exhibit variations in speed due to overall hardware configuration;
>   3. Processing time is highly dependent on the number of frames.
>
>   Therefore, we report a performance range obtained from multiple runs of the baseline methods, rather than a specific runtime. For our own method, we conduct more extensive testing (about 10 runs) under different environments to obtain a more accurate runtime reported in the table. Our intention is to show:
>     1. STAR (averaging 15.8 seconds per video with 30.2 sampled frames) operates in the same time scale as models like GPT-4o (0–30 seconds for 32 sampled frames).
>     2. STAR achieves comparable accuracy to GPT-4o (>10 minutes for 384 sample frames) but completes processing much faster.
>
> + Specifically, we implement 3 variants of the frame selector for different computational resources.
>
>   1. **Vanilla version**: The frame selector is an LLM that takes as input the IDs and textual descriptions of all visible frames, such as captions generated by an image captioner. It outputs the selected frame IDs. The prompt is directly instructing the LLM to select relevant frames.
>   2. **AKeyS-inspired version**: This variant shares the same input and output format as the Vanilla version but employs specially designed prompts to encourage the LLM to consider task goals and scene transition.
>   3. **T\*-inspired version**: In this case, the frame selector is an MLLM, and its input additionally includes the actual images of visible frames. These images are further arranged into an image grid to facilitate selection. The output remains the same.
>
>   The selected frame IDs produced by the frame selector are subsequently added to the *Visible Frame Dictionary*. It is important to note that the *frame selector* and the *video trimmer* are two distinct and independent temporal tools, both scheduled by the LLM planner. We will revise and highlight this point in the later version.
>
> ## Weakness 2
>
> > The performance with more frames
>
> We conducted experiments to evaluate the impact of increased frame sampling rates. Due to API budget and time constraints, we randomly sampled 1,000 examples from the Video-MME test set for this evaluation. The accuracy results are as follows. They demonstrates that the STAR framework continues to improve performance as the number of sampled frames increases. For example, under a denser sampling setting (1 fps, up to 384 frames), the framework achieves a **5.2% accuracy gain**.
>
> | Model  | LLM Planner | Average Frames | Short (%) | Medium (%) | Long (%) | All (%)           |
> | ------ | ----------- | -------------- | --------- | ---------- | -------- | ----------------- |
> | GPT-4o | -           | 32             | 68.6      | 60.6       | 56.0     | 61.5              |
> | GPT-4o | -           | 100            | 72.8      | 63.2       | 59.5     | 64.9              |
> | GPT-4o | -           | 1 fps / 384    | 79.2      | 70.4       | 66.5     | 71.8              |
> | STAR   | GPT-4o      | 31.3           | 78.2      | 68.7       | 62.7     | 69.6 (8.1% ↑)     |
> | STAR   | GPT-4o      | 100.2          | 80.1      | 71.0       | 66.4     | 72.4 (7.5% ↑)     |
> | STAR   | GPT-4o      | 0.98 fps / 384 | **85.3**  | **78.0**   | **68.5** | **77.0** (5.2% ↑) |
>
> ## Weakness 3
>
> > How STAR solves the "unbalanced number and diversity of tools" issue? How to balance tools within category?
>
>  The STAR framework addresses the issue of unbalanced tool quantity and diversity primarily by **removing the tools that the LLM Planner tends to over-rely on** under the no-constraints setting (e.g., the Video QA Tool and Image QA Tool). With these tools no longer “monopolizing” the calls, the distribution of tool usage becomes more balanced both across and within tool categories.
>
> **Below are some supporting data:**
>
> Below is the distribution of tool usage frequencies on the Video-MME test set for both the **No Constraints setting** and the **STAR framework**. The **No Constraints setting** setting is described in line 253-255 in the paper.
>
> | Tool Type & Name                          | No Constraints (%) | STAR framework (%) |
> | ----------------------------------------- | -------------- | -------------- |
> | **Temporal Tools All**                    | **32.1**       | **35.7**       |
> | Frame Selector                            | 27.1           | 21.3           |
> | Temporal Grounding                        | 2.1            | 8.2            |
> | Temporal Referring                        | 0.0            | 1.4            |
> | Video Trimmer                             | 1.3            | 2.6            |
> | Action Localization                       | 1.6            | 2.2            |
> | **Spatial Tools All**                     | **23.6**       | **33.1**       |
> | Object Detector                           | 2.3            | 10.2           |
> | Bbox Marker                               | 0.2            | 1.3            |
> | Image Captioner                           | 2.1            | 7.9            |
> | Image QA                                  | 17.8           | 5.6            |
> | Text Detector                             | 1.0            | 2.5            |
> | Relevant Patch Zoomer                     | 0.2            | 3.7            |
> | Semantic Segmentation                     | 0.0            | 1.9            |
> | **Both (Temporal and Spatial Tools) All** | **5.4**        | **16.1**       |
> | Google Search                             | 0.3            | 1.3            |
> | Object Identifier                         | 1.0            | 3.6            |
> | Action Recognition                        | 0.5            | 1.4            |
> | Image Grid QA                             | 0.1            | 5.7            |
> | Multiple Image QA                         | 2.8            | 1.2            |
> | Python_Code_Generator                     | 0.0            | 1.4            |
> | Object Tracker                            | 0.7            | 1.5            |
> | **General Tools All**                     | **38.9**       | **15.1**       |
> | Text Summarizer                           | 2.2            | 8.3            |
> | Video Summarizer                          | 3.5            | 2.1            |
> | Video QA                                  | 33.2           | 4.7            |
>
> From this raw data, we can compute the **variance of tool usage counts within each tool category**：
>
> | Tool Type      | Var. *of* No Constraints | Var. *of* STAR framework |
> | -------------- | ------------------------ | ------------------------ |
> | Temporal Tools | 134.26                   | 69.90 (64.36 ↓)                   |
> | Spatial Tools  | 41.34                    | 11.09 (30.25 ↓)                   |
> | Both           | 0.92                     | 2.95 (2.03 ↑)                    |
> | General Tools  | 307.45                   | 9.69 (297.76 ↓)                    |
>
> **It can be observed that after adopting the STAR framework, the variance of tool usage counts within each category generally decreases, indicating a more balanced utilization of tools within each class.**
>
> In future work, we plan to further explore how to achieve balanced tool usage within the same category, potentially by manually annotating the ground-truth toolchains for each problem. This would allow us to analyze the ideal distribution of tool usage within categories and design appropriate constraints or incentives to guide the LLM planner toward balanced usage.
>
> ## Weakness 4
>
> > Missing failure cases
>
> **We thank the reviewer for pointing out this limitation. In future versions, we will include representative failure cases along with an analysis of their underlying causes.**
>
> The main types of failure cases are as follows:
>
> 1. **Missing or ambiguous visual information.**
>    In some cases, the video alone does not provide sufficient information and must be supplemented with subtitles or audio cues, e.g., Video-MME test set 302-1. We plan to address such issues by incorporating subtitle inputs and developing tools for audio understanding.
> 2. **Incomplete understanding of the video's main theme.**
>    Our STAR framework sometimes fails to fully capture the main theme of a video due to sparse frame sampling. We aim to improve performance on such global reasoning tasks by encouraging denser sampling in future versions.
> 3. **Difficulty in inferring underlying motivations behind human actions.**
>    STAR struggles with questions that require understanding the deeper intent behind actions. For example, Video-MME test set 312-2: “Why does Michael Bierut mention religious symbols?” This suggests that capturing fine-grained causal reasoning remains a fundamental difficulty in video understanding.

---

> > ### Comment · Reviewer_QhNP · 2025-08-06
> >
> > Thank you to the authors for their detailed rebuttal.
> >
> > I have carefully read the response and revisited the paper and original reviews. My main concerns are well addressed, especially for the missing implementation details and the performance of the proposed method with more frames. Therefore, I will raise my final rating from 3 to 4.

---

> > > ### Author Response · Authors · 2025-08-06
> > >
> > > We sincerely thank you for carefully revisiting our paper and for the thoughtful reconsideration of the rating. We appreciate your constructive feedback, which helped improve the clarity and completeness of our work.

---

### Note · Authors · 2025-08-13

We sincerely thank all reviewers for their constructive feedback and recognition of our work. Reviewers consistently found our paper clearly written, and acknowledged that our proposed STAR framework is effective—achieving superior accuracy, frame efficiency, and runtime performance across multiple datasets and settings. They also recognized the well-structured design and categorization of Video Toolkit, the novelty of our interleaved tool scheduling strategy, and the robustness and portability of our STAR framework.

In the rebuttal process, we provided additional experiments to further validate STAR framework's performance:

1. **Scaling with sampling rate** – STAR’s performance improves as the video sampling rate increases, demonstrating scalability with higher information density.
2. **Generalizability across base LLMs** – STAR consistently delivers strong results when paired with different base LLMs, confirming its robustness and adaptability.

We also addressed the reviewers’ questions in detail. Specifically, in response to reviewer **QhNP**, we used variance statistics to demonstrate that tool usage within the same category tends to be balanced, and we analyzed and summarized common failure cases. For reviewer **JAXG**, we provided a more detailed comparison between our work and related methods (e.g., DoramonGPT and the query-decompose family), and offered a more rigorous definition of the toolchain shortcut. In response to reviewers **ah2Q** and **Fvz6**, we clarified how factors such as sampling rate and the use of subtitles influence performance in video question answering, provided data sources to ensure a fair comparison, and confirmed the strong performance of our approach. We also clarified several implementation details, such as the mechanism of the frame selector, as well as experimental settings, such as the runtime measurement protocol.

To summarize, in this work, we augment LLM with an extensible video toolkit and propose STAR, a spatiotemporal reasoning and toolcalling framework , enabling efficient video understanding. During the rebuttal process, we engaged in constructive discussions with reviewers, addressed their concerns, and received further recognition of our work. We sincerely thank the reviewers and Area Chair for their time and effort. We respectfully hope that the Area Chair will recognize the novelty and significance of our contributions and consider our work for acceptance at NeurIPS’25.

---

### Decision · Program_Chairs · 2025-09-17

**Decision:**

Accept (poster)

**Comment:**

The introduces the Spatiotemporal Reasoning Framework (STAR) and an accompanying Video Toolkit to enhance the video question answering capabilities of Multimodal Large Language Models (MLLMs), demonstrating strong performance gains.

The initial reviews were positive but with notable concerns regarding implementation details, generalizability, and the definition of a "Toolchain Shortcut". In a comprehensive rebuttal, the authors successfully addressed these issues by providing detailed explanations of their methodology, presenting new experimental results showing the framework's scalability with more frames and its portability to other MLLMs like Gemini and Qwen. The AC agrees with the reviewers' opinions (4/5/4/5) and recommends acceptance.